# Multi-annual predictions of hot, dry and hot-dry compound extremes

Alvise Aranyossy[1,2], Paolo De Luca[1], Carlos Delgado-Torres[1], Balakrishnan Solaraju-Murali[1,3], Margarida Samso Cabre[1], and Markus G. Donat[1,4]

[1]Barcelona Supercomputing Center (BSC), Barcelona, Spain
[2]Universitat de Barcelona, Barcelona, Spain
[3]Zurich Insurance, Via Augusta 200, 08021, Barcelona
[4]Institució Catalana de Recerca i Estudis Avançats (ICREA), Barcelona, Spain

**Correspondence:** Alvise Aranyossy (alvise.aranyossy@bsc.es)

**Abstract.** Hot-dry compound extremes have recently gained increasing attention due to their potential impacts on environments and societies. For these reasons, assessing climate predictions is essential to providing reliable information on such extremes. However, despite several studies focusing on compound extremes in the past and climate projections, little is known on a multi-annual timescale. In this regard, decadal climate predictions have been produced to provide useful information for this specific timescale. Thus, we evaluate the ability of the CMIP6 multi-model decadal climate hindcast to predict hot-dry climate extremes, as well as their hot and dry univariate counterparts, for the forecast years 2-5. The multi-model skillfully predicts hot-dry compound extremes and hot extremes over most land regions, while the skill is more limited for dry extremes. However, we find only minor and spatially limited improvements from the initialisation of the hindcasts, especially for the hot-dry compound extremes, with most of the skill coming from external forcings, especially long-term trends. Finally, we find that the decadal hindcast is able to reproduce the connections between the compound extremes and their hot and dry univariate components. Evaluations of decadal hindcasts, such as this, are an essential tool for establishing the potential and limitations of these products. In turn, they represent a necessary step in providing reliable and valuable information regarding such impactful extremes.

## 1 Introduction

Extreme weather events, such as droughts, heatwaves, heavy precipitation and storms, often result from a combination of different physical processes that can interact on various spatial and temporal scales. The aggregation or sequence of these extreme events on the same or different temporal and spatial scales can be considered a compound extreme (Bevacqua et al., 2021). The combination of these hazards can amplify the impacts on climate-vulnerable sectors and society. However, analyzing them solely from a univariate perspective may lead to a severe underestimation of the risk (Leonard et al., 2014; Zscheischler and Fischer, 2020). Furthermore, the dependence among the univariate variables of a compound extreme can increase the likelihood of such events, compared to situations where the univariate variables show no dependence (Zscheischler and Seneviratne, 2017).

Among these compound extremes, hot and dry compound extremes have gained increasing attention due to the potential for elevated environmental, economic, and societal losses. In July and August 2010, Russia experienced strong dry and hot conditions, resulting in massive wildfires and a loss of 20%-30% of the grain production (Grumm, 2011). Other examples from recent years include the European summers of 2003, 2015, and 2018, all characterized by record-breaking hot and dry conditions (Sutanto et al., 2020; Vogel et al., 2020; Zscheischler and Fischer, 2020). Overall, the beginning of the 21st century has witnessed a rise in the frequency of droughts and heatwaves, leading to the belief that this trend will continue throughout the century (Ionita et al., 2017). Several studies have already focused on the variability of these extreme events in the past and how they are projected to change in the future (Zscheischler et al., 2018; Ridder et al., 2021; Bevacqua et al., 2022; De Luca and Donat, 2023). However, little is known about the predictability of these events on multi-annual time scales. Trustworthy predictions are crucial for developing strategic plans to mitigate potential impacts. From this perspective, predictions that cover a multi-annual range could be more useful for potential users than projections for the long-term future. For instance, multi-annual predictions of compound hot-dry extremes can inform strategic and preventive decisions across sectors. For instance, they can support agricultural planning such as irrigation investments, the multiplication of drought-resilient crop varieties, post-harvest management, and enhance the preparedness against pests and diseases (Delgado-Torres et al., 2025). Multi-annual climate information can also guide infrastructure and energy planning (Dunstone et al., 2022), including the enhancement of energy storage capacity or reinforcement of urban green areas to mitigate heat and dry stress, and anticipate impacts on society and environment.

Decadal predictions bridge the gap between short-term seasonal predictions and long-term climate projections. Unlike climate projections, climate predictions prescribe the initial state of the climate system to closely represent the observed climate state at the time of the prediction's initiation. This process is called model initialisation (Meehl et al., 2021). To account for the observational uncertainty in the initial conditions, these climate predictions are initialised from a certain number of slightly perturbed states. Each of the outcoming simulations is called an ensemble member. Decadal prediction systems initially contributed to the Coupled Model Intercomparison Project Phase 5 (CMIP5; Taylor et al. (2012)) and Phase 6 (CMIP6; Eyring et al. (2016)) and have been shown to produce skilful predictions for several climate variables such as near-surface temperature and, to a lesser extent, precipitation (Smith et al., 2019; Delgado-Torres et al., 2022). Multi-annual predictions, which focus on the initial period of the decadal predictions (2-5 years), demonstrate the potential to provide helpful information for various socio-economic sectors. These predictions demonstrate significant skill for the average of forecast years 2-5, which in some cases is comparable, especially in the prediction of temperatures, to the skill shown for seasonal predictions in the average of the corresponding forecast months (Kushnir et al., 2019; O'Kane et al., 2023; Delgado-Torres et al., 2023). Further studies have also investigated the predictability of extreme events on a multi-annual scale, providing comparable results: temperature-related extremes show stronger prediction skill compared to precipitation-related extremes (Solaraju-Murali et al., 2019; Delgado-Torres et al., 2023). To the best of our knowledge, decadal predictions have only been assessed for climate variables or univariate climate extremes. Here, we aim to assess the skill of hot-dry compound extremes in multi-annual predictions within a multi-model setup. We also investigate the improvement of the skill due to the initialisation of the forecast.

Finally, given the importance of the interconnections between compound extremes and their univariate counterparts, we also analyse these correlations and compare the results between observations and models.

## 2 Data and Methods

We analyse the prediction skill of hot-dry compound extremes for a multi-model ensemble from the Coupled Model Inter-comparison Project Phase 6 (CMIP6) (Eyring et al., 2016). The list of the models, the number of ensemble members and the corresponding horizontal resolutions are presented in Table A1. For each model, we use two different sets of experiments. The first set, hereafter referred to as DCPP MME, consists of 10-year initialised hindcasts from the Decadal Climate Prediction Project (DCPP) (Boer et al., 2016). The DCPP, being a multi-system approach, does not specify standard data or methods regarding the initialisation process. For this reason, for any further information, we refer to the references in Table A1. This set of retrospective decadal predictions is produced every year from 1960 to 2009. For this study, we select and analyse the forecast years 2-5 from these decadal forecasts. Thus, in this specific setup, the initial forecast-year 2 is 1962, while the last forecast-year 5 is 2014. We select this subset from the decadal forecast because previous studies have shown that this time-frame benefits from the impact of initialization, thus providing added information compared to uninitialized simulations (Boer et al., 2016). The second set of experiments, referred to hereafter as Hist MME, consists of CMIP6 historical forcing simulations from 1962 to 2014. To estimate the skill of DCPP MME, we use two observation-based reference datasets. The first reference dataset, which will be referred to hereafter as GPCC-BEST, combines the Global Precipitation Climatology Centre (GPCC, Schneider et al. (2016)) for monthly total precipitation and the Berkeley Earth Surface Temperatures (BEST, Rohde and Hausfather (2020)) for daily maximum and minimum temperatures. However, observational datasets are influenced by the presence of missing data. To overcome this problem, and provide a second benchmark for our analyses, we use a second reference dataset that combines daily total precipitation, daily maximum and daily minimum temperature and is obtained from the European Centre for Medium-Range Weather Forecasts (ECMWF) Reanalysis v5, and will be referred to as ERA5 (Hersbach et al., 2020).

### 2.1 Univariate and compound extreme indicators

To identify hot-dry compound extreme days, we first define univariate hot and dry extremes. For temperature extremes, named hereafter TX90p, we select the days above the 90th percentile of daily maximum temperature, measuring the frequency of unusually warm days in each year, as per the definition of the Expert Team on Climate Change Detection and Indices (ETCCDI; Zhang et al. (2011)). The percentile is calculated with a distribution every day of the year, and with a 5-day running window. The use of a 5-day window enables the avoidance of seasonal biases associated with more extended window periods (Brunner and Voigt, 2024). For DCPP MME, this process is lead-time dependent; thus, we select every lead-day of the predictions (with its 5-day window) to build its corresponding distribution for that specific day. This method implicitly accounts for the drift of the decadal predictions. The percentile distributions are based on the entire period of the study, which, in the case of Hist MME, GPCC-BEST and ERA5 is the timeframe 1962-2014, while for DCPP MME, it depends on the forecast year, going

from 1962-2011 for forecast year 2 to 1965-2014 for forecast year 5. In the case of the DCPP MME and Hist MME, the percentile is calculated using data from all the ensemble members for each model individually to account for the models' different variances.

For dry extremes, we use two different indicators: the Standardized Precipitation Index (SPI, McKee et al. (1993)) and the Standardized Precipitation-Evapotranspiration Index (SPEI, Vicente-Serrano et al. (2010)). The use of these two indicators allows us to explore two different types of drought. With the SPI, we explore a drought based only on precipitation (meteorological drought). In contrast, with the SPEI, we explore droughts based on the combination of precipitation and following evapotranspiration (hydrological droughts). These two indicators are calculated using a two-step method: an accumulation process and a standardization process. For the standardization process, we use a gamma distribution for the SPI and a log-logistic distribution for the SPEI, the latter utilising unbiased probability-weighted moments (Beguería et al., 2014; Stagge et al., 2015). The result is a standardized index for total precipitation (SPI) and one for the difference between precipitation and potential evapotranspiration (SPEI), each computed during selected accumulated periods, with potential evapotranspiration (PET) calculated using the Hargreaves method (Hargreaves, 1994). This specific method, which utilises precipitation, minimum and maximum temperatures, provides a good representation of potential evapotranspiration using relatively easy-to-access variables and minimal computational effort (De Luca and Donat, 2023). For computing these indices DCPP MME, we follow Solaraju-Murali et al. (2019). This method treats the accumulation and standardization steps differently due to the nature of decadal predictions. First, the accumulation is done from the first to the last forecast month of every initialisation. In the second step, the standardization is performed individually by building a distribution for each lead-month, using all initialisations together. As for TX90P, the lead-time dependent standardization process accounts for the drift in decadal predictions. Based on the results for the ensemble mean, the distributions show differences between lead years of about 5%. The results for the single models show higher differences, but still considerably smaller compared to the absolute values obtained from the distributions, and limited to specific regions (not shown here). From the resulting time series (one for each month), we form the distribution used to calculate SPI and SPEI. Consequently, the start and end of the reference period will depend on the forecast month analysed. This process is done separately for each ensemble member. In this study, we examine a 3-month accumulation period. Thus, we analyse SPI3 and SPEI3, which are indicators of meteorological (SPI) and hydrological (SPEI) wet (positive values) or dry (negative values) conditions. Finally, at a grid-point level, we define as months with drought conditions all months with SPI or SPEI <= -1 and name these dry indices as SPI3dry and SPEI3dry, respectively, where 3 represents the accumulation period in months (De Luca and Donat, 2023).

Hot-dry compound extremes are then identified following De Luca and Donat (2023), which retain all those TX90p days which fall into SPI3dry or SPEI3dry months. With this method, we obtain two types of hot-dry compound extremes: one based on SPI (HDSPI3, Hot-Dry selected by SPI3dry) and one on SPEI (HDSPEI3, Hot-Dry selected by SPEI3dry).

To assess the robustness of the extremes' thresholds selected in this study, we perform the same analysis of this study for hot extremes over the 95th percentile, SPI and SPEI with dry conditions below -1.5, and hot-dry compound extremes computed over these two previous conditions. In addition, different accumulation periods for the dry conditions represent different types of droughts. For this reason, considering the importance of assessing the influence of different types of droughts on hot-dry

compound extremes, we also perform a sensitivity test for dry and hot-dry compound extremes for accumulation periods of 6
and 12 months.

## 2.2 Forecast quality assessment

We use the Spearman's rank correlation coefficient to assess the skill of the DCPP MME mean in predicting the annual TX90p,
HDSPI3 and HDSPEI3, as well as the SPI3dry and SPEI3dry for the average of the forecast years 2-5. Even if the computation
of the indicators is done at an ensemble-member level, we use the ensemble mean as in climate predictions, the ensemble
mean is typically interpreted as the predictable component of the signal (Eade et al., 2014; Smith et al., 2019). To compare
the reference datasets to the forecast average, we perform a running mean equal to the forecast years taken into account. This
correlation coefficient measures the relationship between the forecasted and observed time series, respectively ranked from
their lowest to their highest value (Wilks, 2011).

We also use the residual correlation to quantify the skill improvement due to initialisation over the skill that comes from the
model responses to external forcing (Smith et al., 2019). This method involves linearly regressing out the ensemble mean
of the Hist MME from the observations and the ensemble mean of the DCPP MME, obtaining the residual values. Finally,
we calculate the correlation between the two residual time series, thereby measuring how much of the observed variability is
captured by the DCPP MME forecast that is not already captured by the Hist MME forecast. To maintain consistency in the
comparison between the DCPP MME and the Hist MME simulations, for the residual correlation analyses, we only use those
models in the DCPP MME ensemble that have at least one member also in the Hist MME ensemble (see Table A1). For all
the correlations shown here, we consider only values statistically significant at the 95% confidence level (p-value $< 0.05$) using
a one-tailed test for the prediction skill and a two-tailed test for the residual skill derived from initialisation, as well as the
analysis on the correlations between univariate and hot-dry compound extremes.

## 3 Results

### 3.1 Multi-annual prediction skill of hot, dry and hot-dry compound extremes

In this section, we present the forecast quality assessment of multi-annual averages for the DCPP MME's forecast years 2-
5. The correlation analysis is performed on the HDSPI3 and HDSPEI3 indicators and their univariate counterparts (TX90p,
SPI3dry and SPEI3dry) between the DCPP MME mean and GPCC-BEST (Figure 1), as well as between DCPP MME mean
and ERA5 as an alternative reference dataset (Appendix Figure S1).

For HDSPI3 (Figure 1a), we observe significantly positive skill in the Mediterranean region, the Balkans, the Arabian Penin-
sula, southern and eastern Asia, the western coast of North America, Central America, most of South America, equatorial
Africa, and southern Australia. On the other hand, HDSPEI3 shows significantly positive skill over most of the globe, with
only the Arctic and sub-Arctic regions, the western Sahara, parts of South America and portions of equatorial Africa showing
non-significant correlations (Figure 1b).

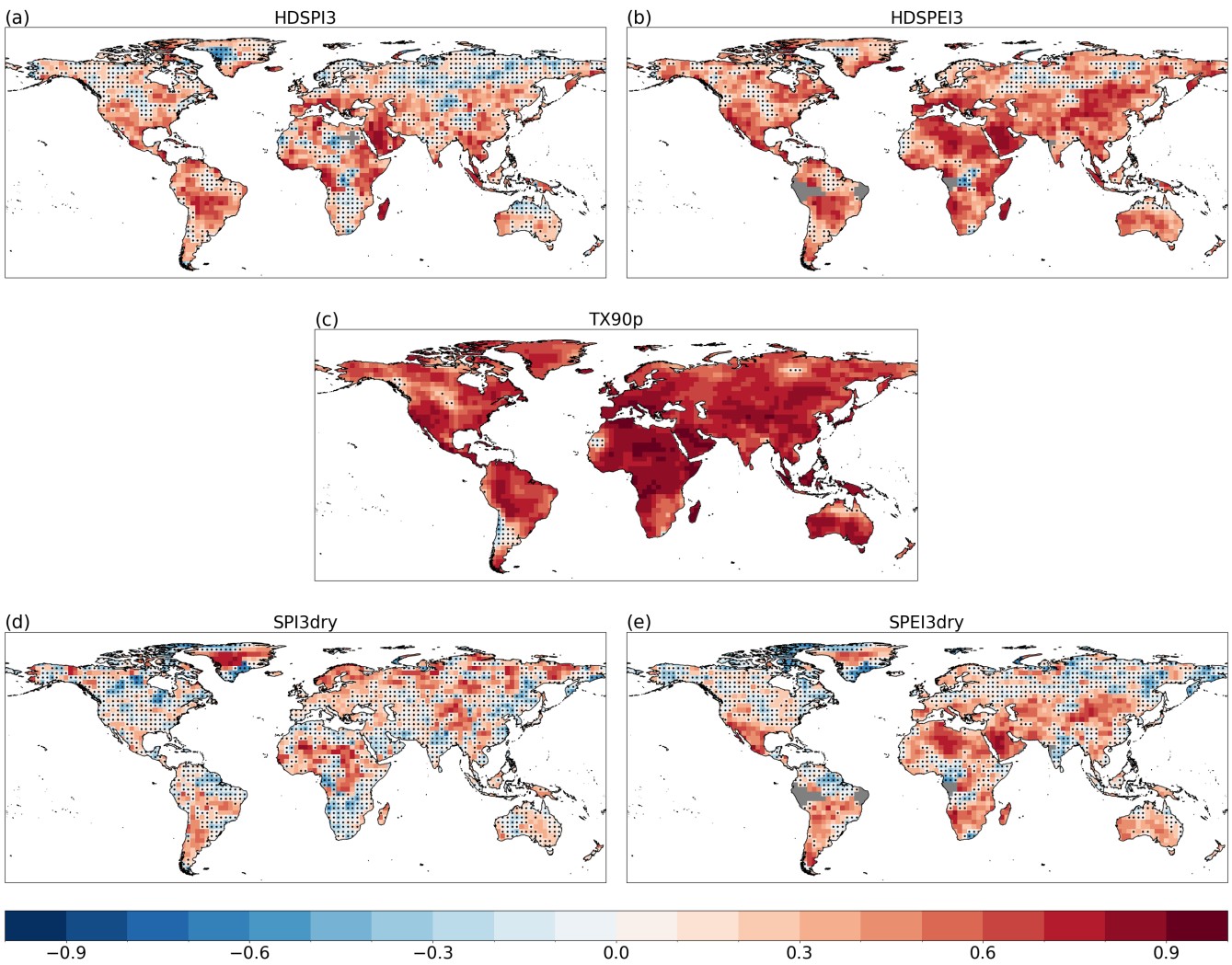

**Figure 1.** Spearman's rank correlation for HDSPI3 (a), HDSPEI3 (b), TX90p (c), SPI3dry (d) and SPEI3dry (e) for the average forecast years 2-5 (1962-2014). Dots indicate regions where the correlation is not statistically significant at the 95% confidence level. Grey areas indicate missing data.

When we analyse the skill of the univariate extremes that contribute to these compound extremes, we find that TX90p (Figure 1c) shows statistically significant positive skill across the globe, with only a few regions, such as the lower latitudes of South America, Western Sahara and some areas of the sub-Arctic regions, showing non-significant values. These results are in accordance with Delgado-Torres et al. (2023).

On the other hand, SPI3dry and SPEI3dry show limited regions with significantly positive skill (Figure 1d,e). For SPI3dry, most of the positive skill is found in northern and equatorial Africa, Scandinavia, certain areas of eastern Europe, and some northern

and central Asian regions. Positive skill is also found in some parts of South America and Australia. Similarly, SPEI3dry exhibits positive and significant correlations over the Mediterranean region, large areas of Africa, the Arabian Peninsula, central and eastern Asia, Central America, South America, and the southern regions of Australia. In general, it appears that SPEI3dry shows higher skill than SPI3dry, especially over the mid-latitudes. If we compare the skill of hot-dry compound extremes to their univariate counterparts, in general, we see an enhanced skill compared to the univariate dry extremes, but not as high as shown by the hot ones. In addition, dry and especially hot-dry compound extremes computed from SPEI show increased skill compared to those calculated with SPI. This characteristic could be partly explained by the contribution of Potential Evapotranspiration, which, relying on temperature, positively contributes to the general skill. This explanation appears to be supported by the fact that the more weight temperature has in the definition of the hot-dry compound extremes (for example, HDSPEI3), the lower skill in dry condition (in this case SPEI3dry) seems to be a limiting factor in the overall skill.

By performing the analysis with ERA5 as the reference dataset (Appendix Figure A1), we obtain some differences among the analysed indices. In the compound extremes (HDSPI3 and HDSPEI3), we find similar areas that show significant skill compared to GPCC-BEST. In addition, as in the previous results for GPCC-BEST, HDSPEI3 shows an enhanced skill compared to HDSPI3. Regarding the univariate extremes, TX90p still exhibits the strongest skill, and SPEI3dry shows stronger skill than SPI3dry, especially in the Northern Hemisphere. In general, compared to GPCC-BEST, the skill using ERA5 as a reference dataset shows a slight diminishment in the area where significant positive correlations are found. This diminishment is especially evident in the dry extremes (SPI3dry and SPEI3dry), where large areas, particularly at high latitudes and in equatorial Africa, show strong negative correlations.

Finally, in the sensitivity test, we find that changing the thresholds of the univariate and compound extremes does not change the overall results (Figure S7). In fact, we find significant correlations in the same areas, with a slight difference between the strength of the correlations, especially in dry and compound extremes. On the other hand, a change in the accumulation period for the dry and compound extremes appears to bring significant differences (Figures S8 and S9). Compared to extremes calculated with a 3-month accumulation, accumulating over 6 and 12 months generally shows a decrease in significant skill, the longer the accumulation period. However, this is not the case with Northeast Asia, where longer accumulations show higher correlations.

## 3.2 Residual correlations and trends

The skill of the decadal forecast shown above (Figure 1 and Appendix Figure A1) can be essentially divided into two components: the skill resulting from external forcings and the skill resulting from initialisation. In particular, the skill coming from the initialisation is beneficial in providing information regarding the variability of such extremes in the upcoming years. In this section, we analyse to what extent the two elements contribute to the overall skill.

To quantify the added skill contributed by the models' initialisation in decadal predictions, we perform the correlation between the residuals of GPCC-BEST and the DCPP MME ensemble mean (Figure 2). The results for HDSPI3 globally show small amounts. Furthermore, these areas are scattered and spatially isolated, not showing a larger region where a clear added skill can be found (Figure 2a). For HDSPEI3, on the other hand, despite showing a similar spatial pattern, areas of consistent added

skill can be identified in central Asia and southern Australia, as well as in some regions of the high latitudes of North America (Figure 2b).

On the other hand, between the univariate extremes, TX90p (Figure 2c) appears to show the most benefit from the initialisation, with added skill located in central and eastern Asia, Greenland and southern Australia. Additionally, we also find some added value in small parts of North America, South America and Africa. For dry extremes, the areas of added skill are more limited.

SPI3dry demonstrates added skill in some small areas of South America, Asia and Scandinavia (Figure 2d). Moreover, northern Africa also shows a homogeneous area of added skill, which, at least in part, corresponds to the significant skill observed for SPI3dry (Figure 1d). On the other hand, SPEI3dry shows slightly fewer areas of added skill, limited to southern Australia and central Asia (Figure 2e), the latter being similar to the region where we find added skill for TX90p.

The residual correlation obtained with ERA5 as a reference dataset yields similar results, both in terms of general patterns and regions showing significant residual correlations (Appendix Figure A2). However, for HDSPI3 and HDSPEI3, as well as in SPI3dry and SPEI3dry, an increase in added skill is noticeable in the tropical regions of South America and eastern Asia. For different thresholds and accumulation, the residual skill yields results similar to those previously shown for the overall skill. Different thresholds lead to identical areas where we observe a significant contribution from initialization, with a slight

reduction in size and strength of the correlation (Figure S10). A change in the accumulation period instead leads to more substantial differences between the 3-month accumulation and the 6- and 12-month accumulations, respectively (Figures S11 and S12). These differences, similar to those in the previous section, appear to intensify as the accumulation period is extended.

In general, the results in Figures 2 and A2 show that, for the univariate hot and dry extremes, we have certain limited areas

where the initialisation significantly adds skill to the forecast (Figure 1, Figure A1). However, the added skill detected in certain areas for the univariate extremes is mostly lost when analysing the compound ones. Indeed, the results suggest that the skill relies strongly on the models' response to external forcings, such as the strong warming trend in temperatures, especially in the case of lower latitudes for HDSPEI3, where we find strong overall skill (in terms of correlation with observations) but little residual correlation.

To better establish the role of external forcings and trends in predicting these extremes, we explore the connection between significant correlations and the signs of the trends (Figure 3). In this analysis, we select the regions of DCPP MME which show significant positive or negative skill. For these specific regions, we then compare the trend signs for DCPP MME and GPCC-BEST, specifically examining whether the trend signs of the two datasets are in agreement. We can then classify regions into four categories: significant positive with a trend in agreement (or disagreement), and significant negative with a trend in

agreement (or disagreement). From these results, a positive significant skill in most regions is tied to an agreement in trend sign between DCPP MME and GPCC-BEST. In addition, in most areas where the correlation is significantly negative, we find that the trends between the two datasets are discordant. An exception can be observed for SPI3dry, where, especially in Eastern Europe, the Western Sahel and regions of mid-latitude South America, the correlations are significantly positive, but the sign of the trend is discordant (Figure 3d). Two factors could help explain this. The first is the weak trend in precipitation in those

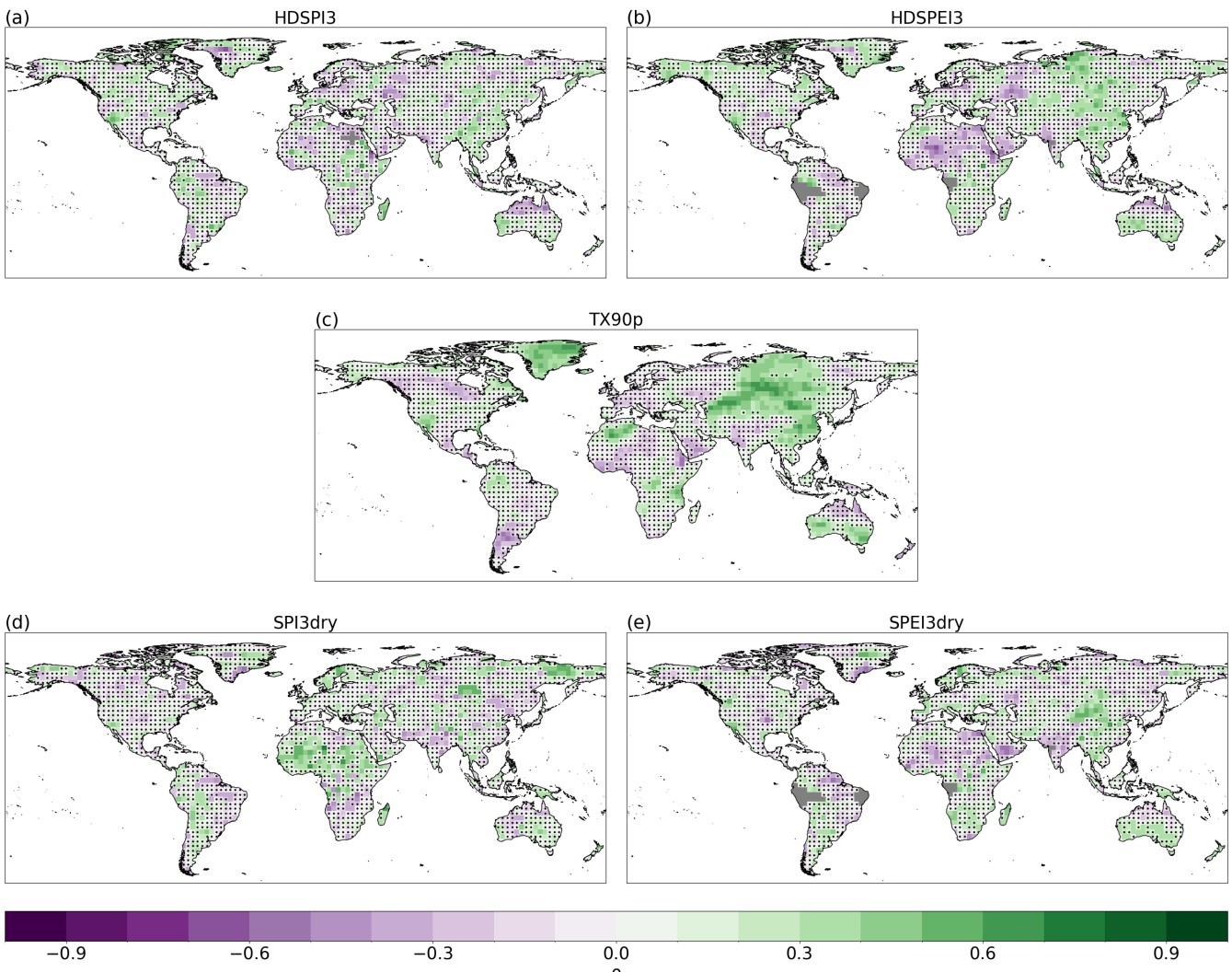

**Figure 2.** Spearman's rank residual correlation for HDSPI3 (a), HDSPEI3 (b), TX90p (c), SPI3dry (d) and SPEI3dry (e) for the average forecast years 2-5. Dots indicate areas where the residual correlation is not statistically significant at the 95% confidence level. Grey areas indicate missing data.

areas for both datasets (Appendix Figure A4), which also explains the opposite scenario, a significantly negative correlation but coinciding trend signs, visible, for example, in South Africa and tropical South America. The second factor is the added skill from initialisation for those grid points, which, despite the discordant trend, leads to a significant positive skill (Figure 2d).

On the contrary, all the extreme indices that include temperature show a strong agreement between positive (negative) signif-

icant correlations and agreeing (discordant) trend signs, specifically evident in HDSPEI3 and SPEI3dry (Figures 3b and 3e).

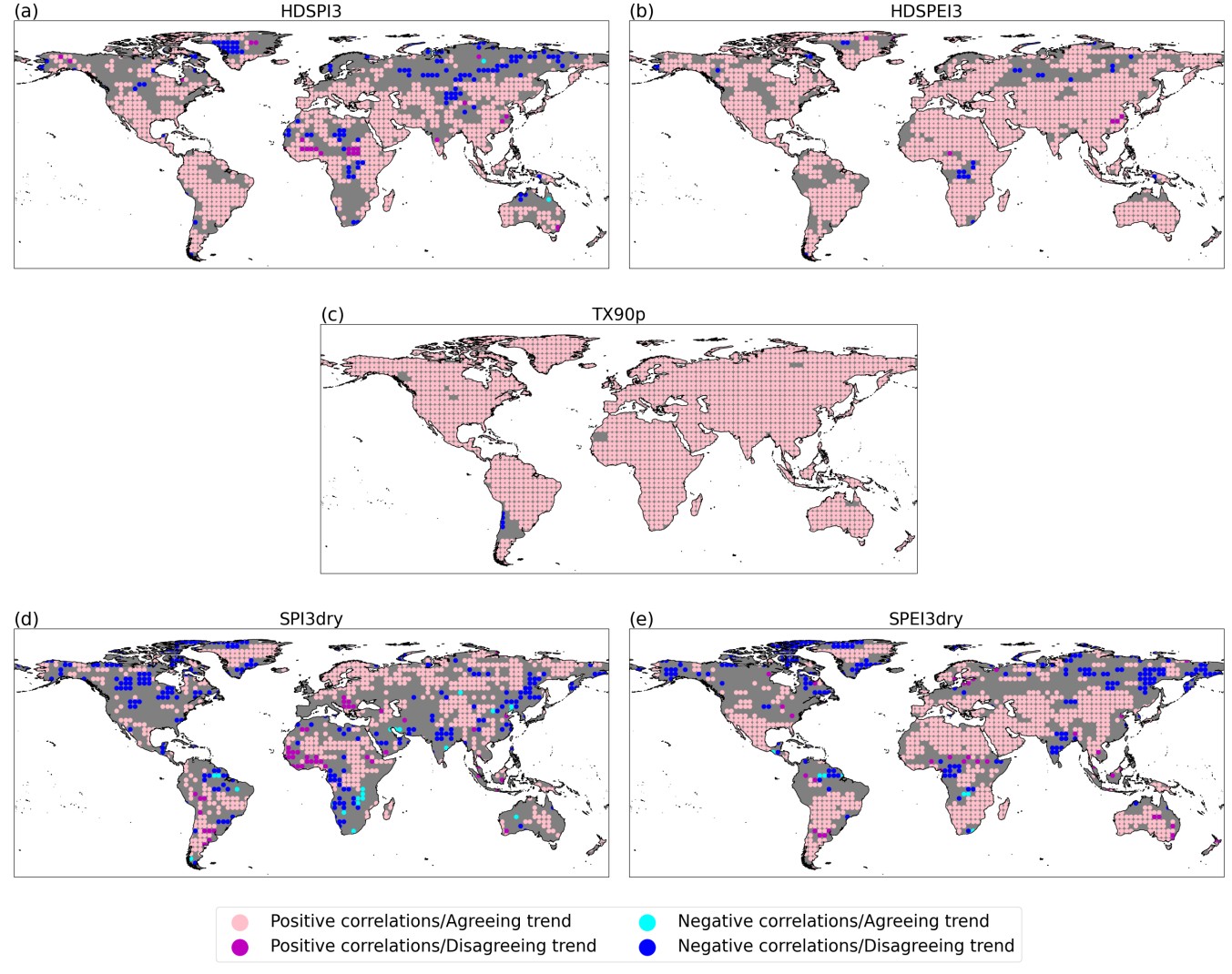

**Figure 3.** Statistically significant Spearman's ank correlation against the trend sign for HDSPI3 (a), HDSPEI3 (b), TX90p (c), SPI3dry (d) and SPEI3dry (e) for the average forecast years 2-5 (as in Figure 1). Pink (magenta) dots indicate areas where the correlation is positive and statistically significant, and where the trends of GPCC-BEST and DCPP MME show the same (opposite) sign. Light blue (Dark blue) dots indicate areas where the correlation is negative and statistically significant, and where the trends of GPCC-BEST and DCPP MME show the same (opposite) sign. Grey areas indicate grid points where the correlation is non-significant.

These results are in line with the findings of Donat et al. (2023). The same conclusions can be drawn concerning the results when using ERA5 as a reference dataset (Appendix Figure A3). Here, the areas with opposite trends and significantly negative correlations are evident in tropical Africa for SPI3dry and SPEI3dry and in the high boreal latitudes for HDSPI3 and HDSPEI3. When examining the trends, we observe that these areas exhibit opposite trends in DCPP MME and ERA5 for the

aforementioned extreme indices. However, in these regions, we can also observe areas with significant negative correlations and agreeing trend signs (Appendix Figure A3d and A3e). These results could be explained by the weak trends in the region (Appendix Figure A5), making it a non-significant factor in determining significant skill. For SPI3dry, we observe the same situation as for GPCC-BEST in tropical South America (Appendix Figure A3d). In this region, in fact, we observe discordant trends between DCPP MME and ERA5, yet significant positive correlations. We can also see, between DCPP MME and ERA5, that in that area, the precipitation trends are not strong (Appendix Figure A5) and that the forecast shows a large region of added skill for ERA5 (Appendix Figure A2d).

In summary, the results discussed above indicate that decadal predictions exhibit little added skill compared to Hist MME, especially for dry and hot-dry compound extremes. However, most of the skill seen in Section 3.1 is still tied to the signal associated with external forcings. In particular, we observe that the presence of the skill, in the absence of added skill from the initialisation, is tied to the trends of the models and the reference datasets, and whether these two trends agree in their signs, especially if the selected variable is affected by strong temperature trends. The results presented in this section highlight the current limitations of multi-annual predictions, particularly in terms of providing information on the upcoming variability of such extreme events. In this regard, the question now lies in the extent to which the different univariate extremes influence the detection and the variability of the compound extremes in the model and the reference datasets.

## 3.3 Correlations between univariate and compound extremes

The method used in this study, which relies on the co-occurrence of hot and dry conditions at the same location, suggests that one of the two univariate extremes may play a more important role in the overall detection of the events. In addition, this co-occurrence may vary across different datasets, for example, between the DCPP MME and the observation-based datasets, resulting in different counts of compound extremes and, consequently, different interannual variations. Thus, we perform a correlation analysis between the annual sum of univariate and compound extremes to assess the representation of the underlying interconnections. Since the HDSPI3 and HDSPEI3 were estimated for every ensemble member, we calculated the correlations between the univariate and compound values for each ensemble member. We then selected the median correlation across DCPP MME. Finally, we consider a significant correlation in those areas where at least 50% of the ensemble members show significant positive or negative correlations.

The results for the correlations between HDSPI3 and HDSPEI3 and the univariate extremes are shown in Figures 4 and 5, respectively. If we look at the correlations between HDSPI3 and TX90p in GPCC-BEST and the DCPP MME (Figure 4a,b), we can see that they both show significantly positive correlations across most of the globe. In addition, the DCPP MME can reproduce the large areas where the connections between HDSPI3 and TX90p are weaker, such as the high latitudes of the Northern Hemisphere. However, it is less effective in areas such as Africa or central Asia, where correlations in DCPP MME are still significant, unlike those in GPCC-BEST. On the other hand, when looking at the correlations between HDSPI3,SPI3dry and SPEI3dry, we see similar patterns (Figures 4c, 4d and 4e 4f, respectively). Both GPCC-BEST and DCPP MME show strong positive correlations between drought occurrence and hot-dry compound extremes in most regions globally, except mainly in northern Africa. However, as seen in the correlations with TX90p (Figures 4a and 4b), DCPP MME partially fails to reproduce

the high spatial differences in connections observed in GPCC-BEST, with only larger areas, such as northern Africa, showing
275 differences in the correlation values (Figures 4d and 4e).

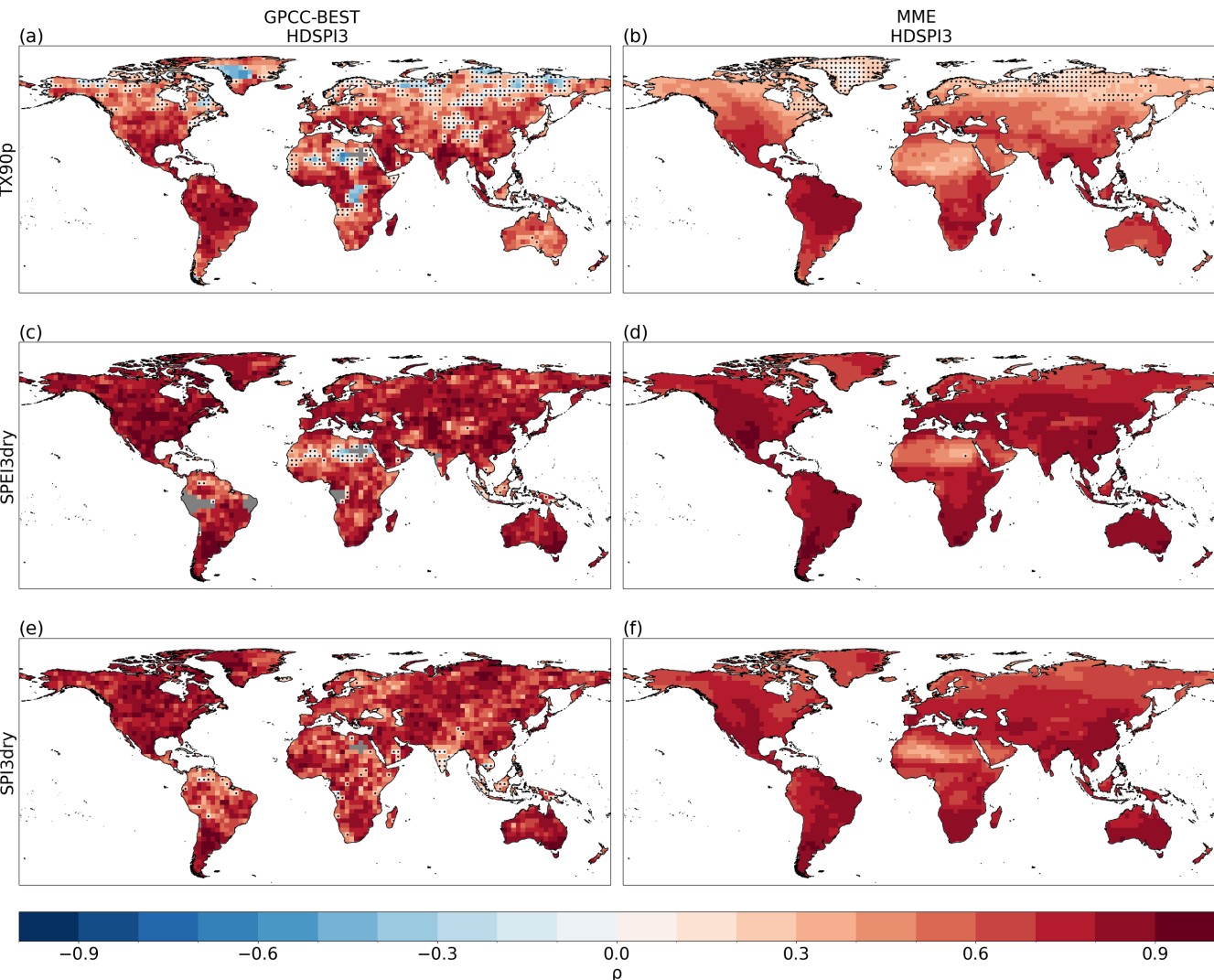

**Figure 4.** Spearman's rank correlation between the annual sum of HDSPI3 and TX90p (a, b), SPEI3dry (c, d) and SPI3dry (e, f), respectively.
For a,c, and e: Correlations are calculated with GPCC-BEST, and dots indicate correlations not statistically significant at the 95% confidence
level. For b, d and f: Correlations represent the median (ensemble-wise) correlation for the DCPP MME ensemble, and dots indicate grid
points where more than 50% of the ensemble members provide correlations not statistically significant at the 95% confidence level. Grey
areas indicate missing data.

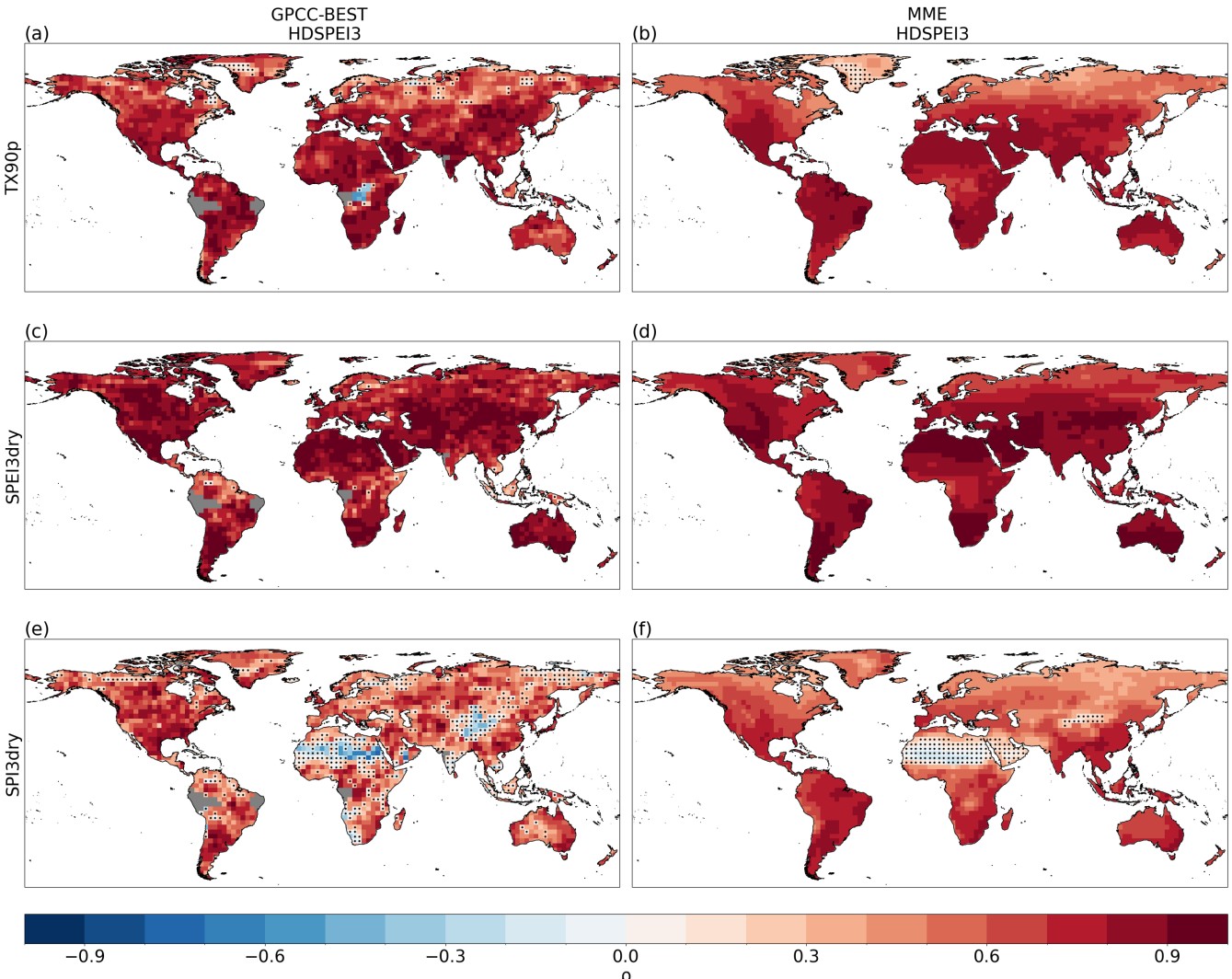

**Figure 5.** Same as Figure 4, but for HDSPEI3.

For the correlations between HDSPEI3 and the univariate extremes (Figure 5), we observe patterns similar to those in Figure 4. In this case, the correlations between HDSPEI3 and TX90p appear more substantial and more widely significant in GPCC-BEST and the DCPP MME, with the high latitudes showing the weakest correlations (Figures 5a, 5b). The correlations between
280  HDSPEI3 and SPEI3dry in GPCC-BEST exhibit strong positive correlations globally, with DCPP MME reproducing them significantly well (Figures 5c, 5d). On the other hand, the correlations between HDSPEI3 and SPI3dry in GPCC-BEST show high spatial differences, with large areas showing non-significant correlations. However, DCPP MME shows non-significant correlations uniquely across northern Africa, the Arabian Peninsula and central Asia (Figures 5e, 5f).

Analysing the correlations between compound and univariate extremes in ERA5, we observe a pattern similar to the results seen above for GPCC-BEST (Appendix Figure A6). In general, both reference datasets (GPCC-BEST and ERA5) show stronger correlations for TX90p with HDSPEI3, especially in the equatorial and mid-latitudes. Still, we find significant correlations between hot-dry compounds and dry extremes, which appear to be an essential driving factor in the variability of both HDSPI3 and HDSPEI3, and more specifically for the dry extreme with which the compound extreme is being calculated (i.e. SPI3dry for HDSPI3, SPEI3dry for HDSPEI3). The DCPP MME can reproduce these general patterns, particularly the weaker correlations in the high Northern Hemisphere latitudes for TX90p and in northern Africa for the dry extremes. However, it fails to represent the high spatial differences in the correlation values, which instead can be seen in the reference datasets. These results can be attributed to the smoothing that inevitably occurs when calculating the ensemble mean, causing the variability of the individual ensemble members to be lost.

## 4 Summary and Conclusions

This study provides an overall assessment of hot, dry and hot-dry compound extremes in multi-annual predictions. The results show that the predictive capabilities of the decadal forecast systems are not only limited to the univariate extremes but also show value for the compound extreme occurrences. Compared to the skill shown in the univariate extremes, compound extremes show higher or lower predictability: global compound skill is smaller compared to hot extremes, especially at high latitudes; on the other hand, they show higher skill compared to their dry univariate counterparts. These differences are likely associated with the fact that temperature-related extremes are more predictable than their precipitation-related counterparts, with the latter's predictability still representing a bottleneck that could damp the signal (Solaraju-Murali et al., 2019; Bellucci et al., 2015; Delgado-Torres et al., 2023; Smith et al., 2019; De Luca and Donat, 2023). From this perspective, the compound extremes partly benefit from the higher predictability of hot extremes.

However, the results from the residual correlation suggest that most of the skill of the compound extremes comes from external forcings, which cause long-term trends in some extreme indices. This conclusion is partially less true for dry extremes, where we can find specific regions with a weak trend, and where the initialisation actively plays a role in providing significant skill. In fact, we observe that in the variables where temperatures play a role in defining the extreme indices (i.e. TX90p, SPEI3dry, HDSPI3 and HDSPEI3), the tie with the trend is nearly ubiquitous (Donat et al., 2023). Thus, a skilful representation of the interannual variability still lacks in multi-annual predictions. This characteristic represents a limit to the usability of these forecasts and the information they can provide.

In addition to these results, the analysis of the interconnections between compound and univariate extremes shows that the decadal forecast systems are generally able to reproduce the connections between the compound extremes and their univariate counterparts based on observations, attributing a leading role to the dry conditions, and specifically to the one on which the compound extreme is calculated (e.g. drought defined based on SPI or SPEI). Thus, if we compare these results with the prediction skill, we can identify where the main limiting factor in the prediction of hot-dry compound extremes may lie: dry extremes show to be the leading factor determining the variability of hot-dry extremes, but on the other hand, it is the dry

extremes that show the least prediction skill on a multi-annual time range.

Additionally, we confirm that using different reference datasets (GPCC-BEST and ERA5) yields results that differ slightly, but not significantly. The differences observed in the two reference datasets, especially for dry and hot-dry compound extremes, could be attributable to the different uncertainties in the datasets, such as the wet bias in Central Africa or the dry bias in the Northern Hemisphere continental areas for ERA5 (Hassler and Lauer, 2021).

Regarding the sensitivity analysis, a change in the threshold of the extreme events does not result in significant differences. The reason could be connected to the threshold of the hot (dry) extremes, which generally belong to the higher (lower) end of the distribution, and typically exhibit the same characteristics, especially in a normal distribution. On the other hand, we observe that a change in the accumulation period has a stronger impact on the skill of predicting dry and hot-dry compound extremes. By increasing the accumulation period of the dry conditions, we are observing different types of droughts and consequent impacts. 3-month accumulation SPI and SPEI are related to more short-term dry conditions, with an impact on stream-flow and soil moisture. On the other hand, 6-month accumulation represents medium-term dry conditions, related to anomalies in specific seasons, with impacts on stream-flow and reservoir levels. Finally, a 12-month accumulation reflects long-term precipitation patterns, with more long-lasting impacts on streamflows, reservoir levels and even groundwater levels (Svoboda et al., 2012). These differences in the type of drought also influence the predictability of such dry conditions. In fact, on one side, selecting a longer accumulation period leads to including a time range which is otherwise not considered, thus changing the nature of the variable in question. On the other side, longer accumulation periods tend to gravitate more towards zero, thus changing the predictability of the indicator. This interpretation could explain the increase and decrease in prediction skill between different accumulation periods in several regions and why the areas where we find the strongest changes between accumulation periods are common to both the overall and residual skills, underscoring the influence of the accumulation period on the internal variability of the dry condition. For this reason, the selection of the accumulation period when calculating SPI and SPEI should be carefully considered, depending on the type of drought one wants to observe.

We conclude that CMIP6 decadal forecast systems can skillfully predict the compound hot and dry extremes over large parts of the world at the multi-annual scale. However, this skill is strongly tied to external forcings, and a significant gap remains for improvement in predicting interannual variability. Finally, the decadal prediction systems can detect the underlying connections between compound and univariate extremes, even if at a large-scale level, showing the leading role of dry conditions. Similar forecast quality assessments should be conducted on a regional and seasonal scale, as well as for other compound indicators, to provide even more insightful and specific information on such extreme phenomena. These further steps could provide the information necessary to enable targeted climate information to support mitigation and adaptation policies against potential damages caused by these events.

*Code and data availability.* Gridded ERA5 data ranging from 1961 to 2014 was downloaded from https://cds.climate.copernicus.eu/. Gridded GPCC data ranging from 1961 to 2014 were downloaded from https://opendata.dwd.de/climate_environment/GPCC/html/fulldata-monthly_

v2020_doi_download.html. Gridded BEST data ranging from 1961 to 2014 were downloaded from http://berkeleyearth.lbl.gov/. CMIP6 dcppA hindcasts and historical simulations were downloaded from https://esg-dn1.nsc.liu.se/projects/esgf-liu. The scripts used to support the results of this analysis are available on GitLab at the following URL: https://earth.bsc.es/gitlab/aaranyos/hotdry_compounds_dcppa.

*Author contributions.*  AA, MD and PDL conceptualized and designed the study. MSC provided the data for the simulations, observations
and reanalysis. CDT and BSM contributed in the analyses. AA wrote the first draft. PDL, CDT and MD contributed to interpreting the results,
discussing the findings and improving the final version of the paper.

*Competing interests.*  The contact author has declared that none of the authors has any competing interests.

*Acknowledgements.*  This research was funded by the Horizon Europe project ASPECT (Grant 101081460). We are also grateful for support
by the Departament de Recerca i Universitats de la Generalitat de Catalunya for the Climate Variability and Change (CVC) Research Group
(Reference: 2021 SGR 00786), and support by the AXA Research Fund. A.A. was funded by MCIN/AEI/10.13039/501100011033 and
ESF Investing in Your Future (Grant PRE2022-104391). P.D.L was funded by Horizon Europe Marie Sklodowska-Curie Actions (Grant
101059659).

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

**Appendix**

**Table A1.** List of models and number of ensemble members used for the DCPP MME and the Hist MME. Abbreviations used for the initialization schemes: atm,atmopheric component; ocn,ocean/sea-ice component; FF, full-field; AN,anomalies; EnKF,Ensemble Kalman Filter; IAU,incremental analysis updating. Where available, the datasets used in the initialization are shown in brackets. For further details we refer to the publications reported as references.

| Forecast systems | DCPP members | Historical members | Spatial resolution | Month initialisation | Initialization scheme | Reference |
|---|---|---|---|---|---|---|
| CMCC-CM2-SR5 | 10 | 1 | 0.9°x1.25° | November | FF atm nudging (ERA-40,Interim,5) and FF ocn EnKF | Nicolì et al. (2023) |
| CanESM5 | 20 | 20 | 2.8°x2.8° | January | FF atm nudging (ERA-Interim) and FF ocn nudging (ORAS5,OISST) | Sospedra-Alfonso et al. (2021) |
| EC-Earth3-i1 | 10 | - | 0.7°x0.7° | November | FF nudging (ERA-40,Interim) and FF ocn nudging (ORAS4) | Bilbao et al. (2020) |
| EC-Earth3-i2 | 5 | - | 0.7°x0.7° | November | FF atmn (ERA-Interim) and AN ocn (ORAS5) | Tian et al. (2021) |
| EC-Earth3-i4 | 10 | 9 | 0.7°x0.7° | November | FF atmn (ERA5) and FF ocn nudging (ORAS5) | Döscher et al. (2021) |
| HadGEM3-GC31-MM | 10 | - | 0.55°x0.83° | November | FF atmn and FF ocn nudging | Sellar et al. (2020) |
| IPSL-CM6A-LR | 10 | 10 | 1.25°x2.5° | January | | Boucher et al. (2020) |
| MIROC6 | 10 | 10 | 1.4°x1.4° | November | FF(JRA55) and AN ocn IAU | Kataoka et al. (2020) |
| MPI-ESM1-2-HR | 10 | 1 | 0.9°x0.9° | November | FF atmn nudging and FF ocn EnKF | Müller et al. (2018) |
| MRI-ESM2-0 | 10 | 10 | 1.125°x1.125° | November | AN ocn | Yukimoto et al. (2019) |
| NorCPM1-i1 | 10 | - | 1.9°x2.5° | October | AN ocn EnKF (clim 1980-2010) | Bethke et al. (2021) |
| NorCPM1-i2 | 10 | 10 | 1.9°x2.5° | October | AN ocn EnKF (clim 1950-2010) | Bethke et al. (2021) |
| MME | 125 | 80 | 2.8°x2.8° | | | |

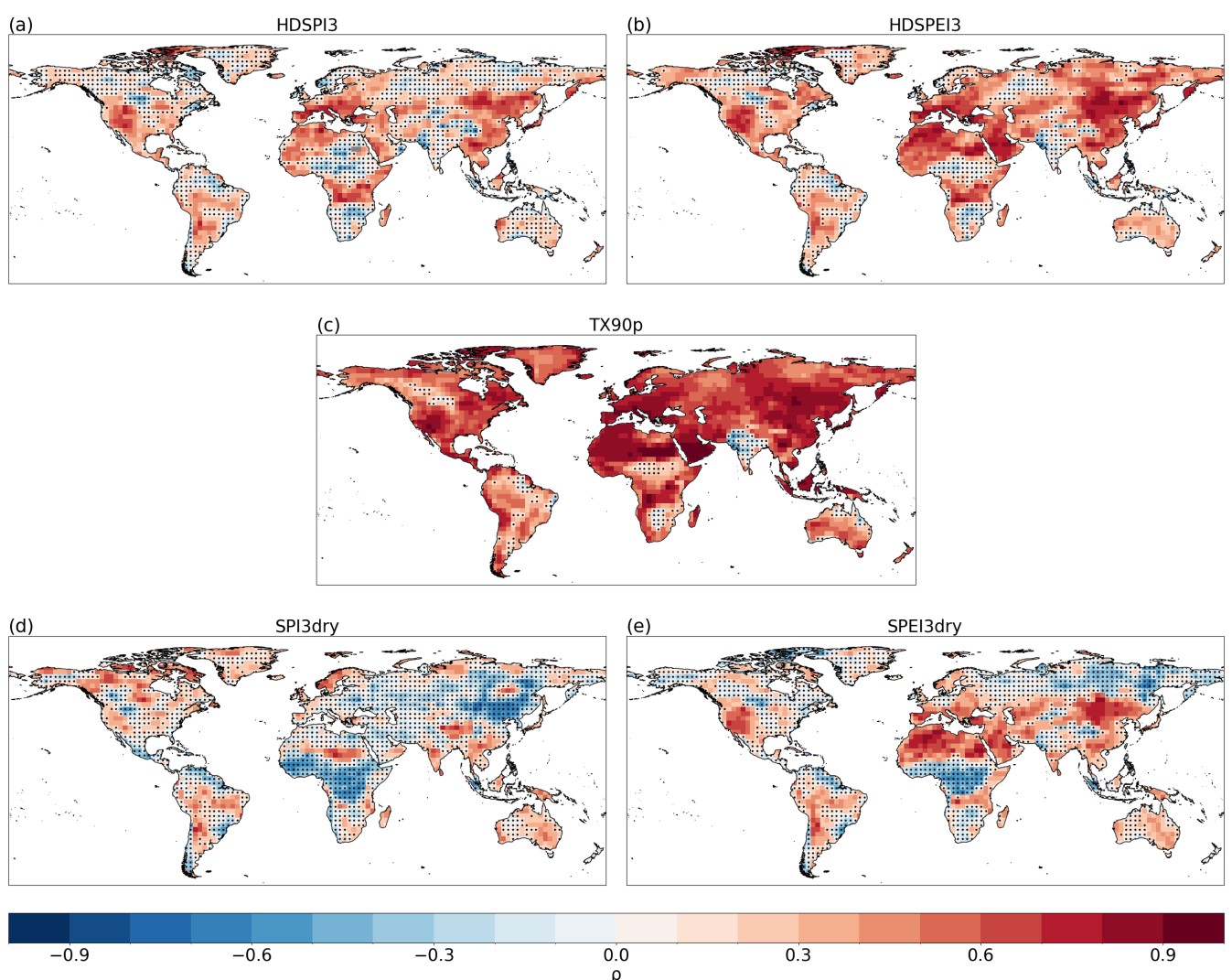

**Figure A1.** Spearman rank correlation for TX90p(a), SPI3dry and SPEI3dry (b,c) and HDSPI3 and HDSPEI3 (d,e) for the average forecast years 2-5. Correlations are performed between the DCPP MME ensemble mean and the ERA5 dataset. Hatchings indicate correlations not statistically significant at the 95% confidence level (p-value < 0.05) using a one-tailed test.

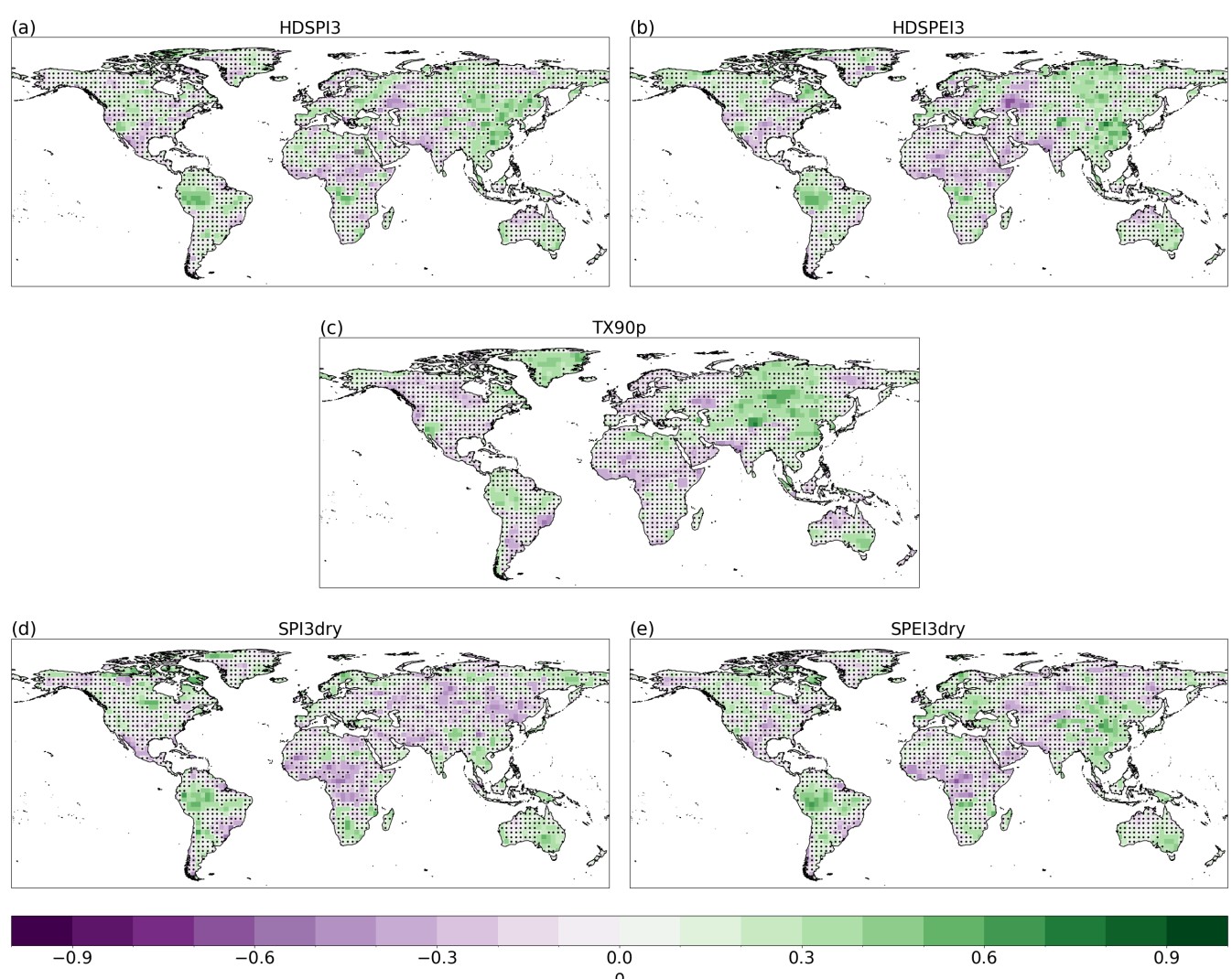

**Figure A2.** Spearman rank residual correlation for TX90p(a), SPI3dry and SPEI3dry (b,c) and HDSPI3 and HDSPEI3 (d,e) for the average forecast years 2-5. Correlations are performed between the DCPP MME ensemble mean and the ERA5 dataset. Hatchings indicate correlations not statistically significant at the 95% confidence level (p-value < 0.05) using a two-tailed test.

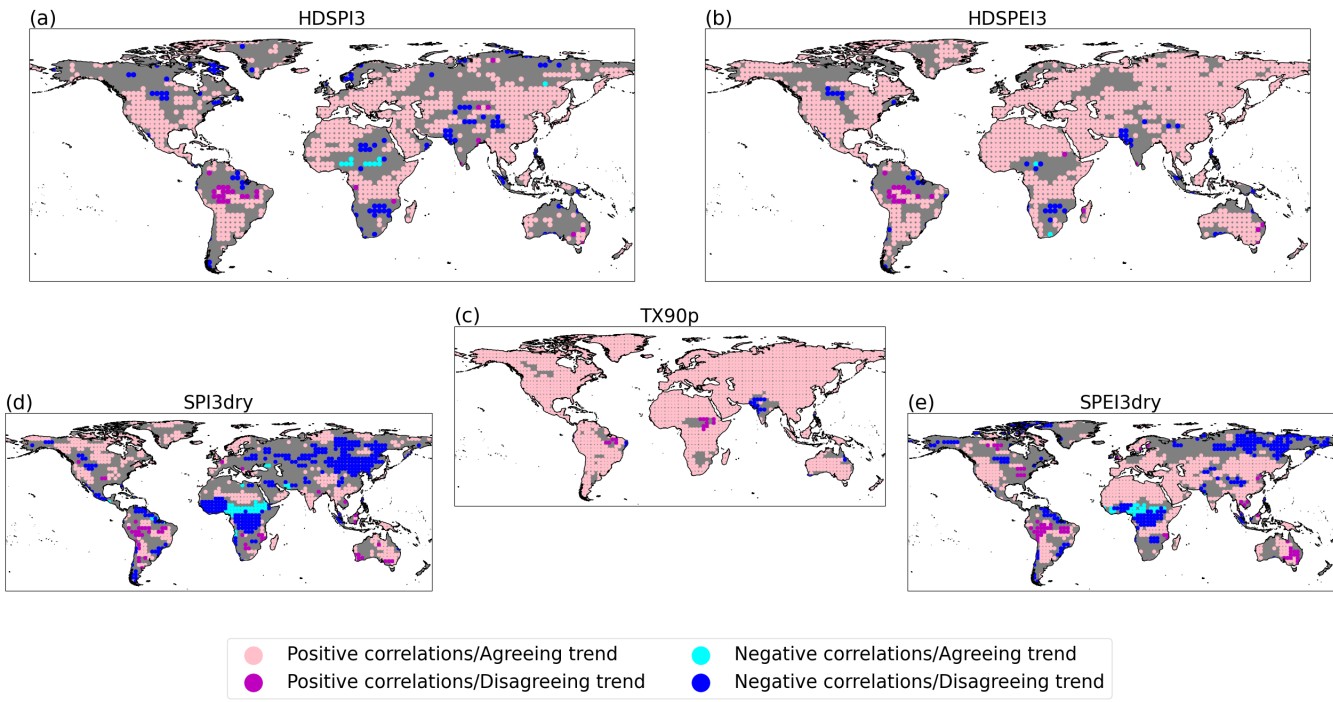

**Figure A3.** Statistically significant Spearman Rank correlations against the trend sign for HDSPI3 (a), HDSPEI3 (b), TX90p (d), SPI3dry (c) and SPEI3dry (e) for the average forecast years 2-5 (as in Figure 1). Correlations and trend sign comparison are performed between the DCPP MME ensemble mean and the ERA5 dataset. Pink (magenta) dots indicate areas where the correlation is positive and statistically significant at the 95% confidence level (p-value < 0.05) and where the trend of GPCC-BEST and DCPP MME show the same (opposite) sign. Light blue (Dark blue) dots indicate areas where the correlation is negative statistically significant at the 95% confidence level (p-value < 0.05) and where the trend of GPCC-BEST and DCPP MME show the same (opposite) sign. Grey areas indicate grid points where the correlation is non-significant.

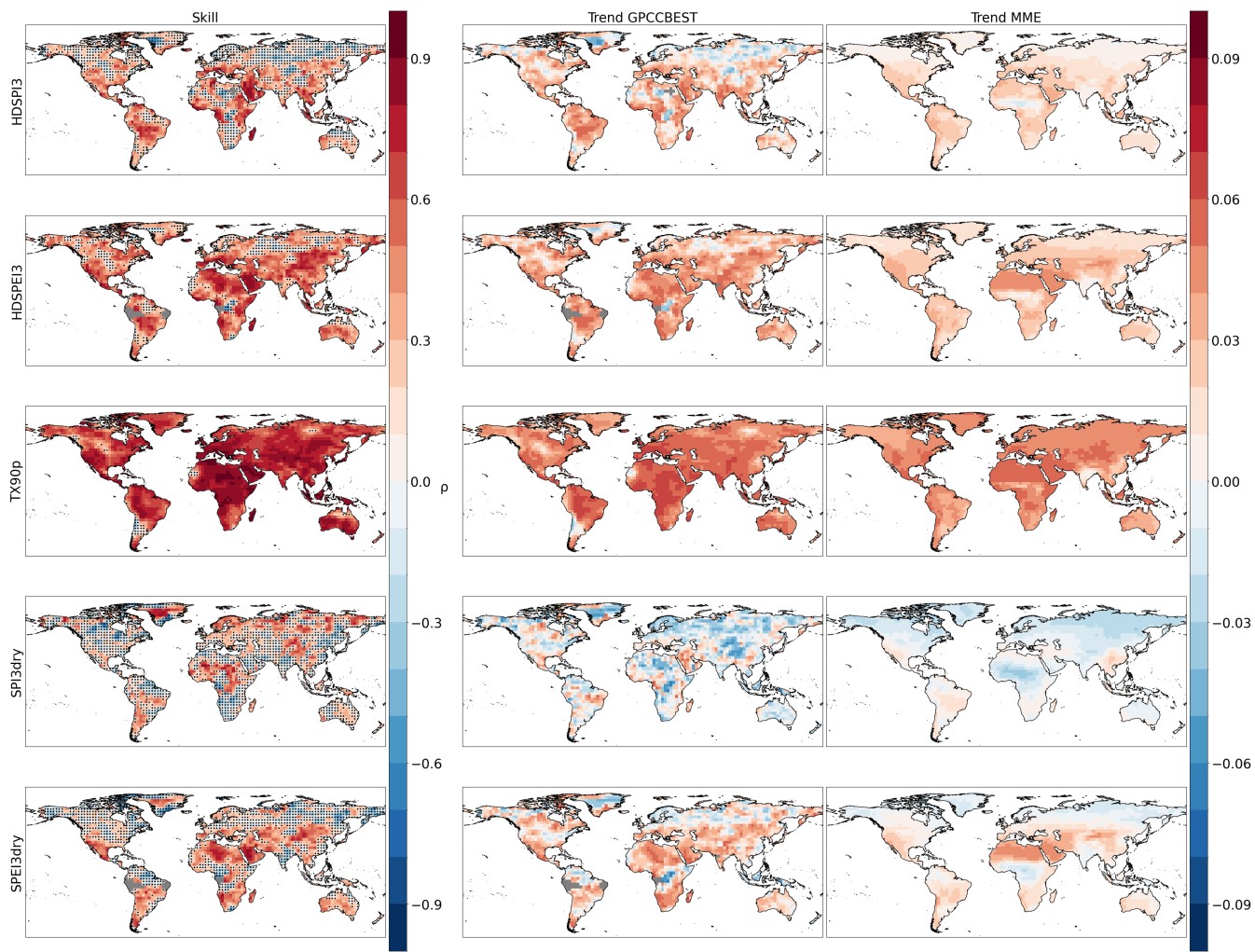

**Figure A4.** First row: Spearman rank correlation between the GPCC-BEST dataset and the DCPP MME ensemble mean for HDSPI3, HDSPEI3, TX90p, SPI3dry and SPEI3dry respectively. Hatchings indicate correlations not statistically significant at the 95% confidence level (p-value < 0.05) using a one-tailed test. Second row: Normalized trends in the GPCC-BEST dataset for HDSPI3, HDSPEI3, TX90p, SPI3dry and SPEI3dry respectively. Third row: Median normalized trends, calculated ensemble-wise, in the DCPP MME ensemble for HDSPI3, HDSPEI3, TX90p, SPI3dry and SPEI3dry respectively. Grey areas indicate missing data.

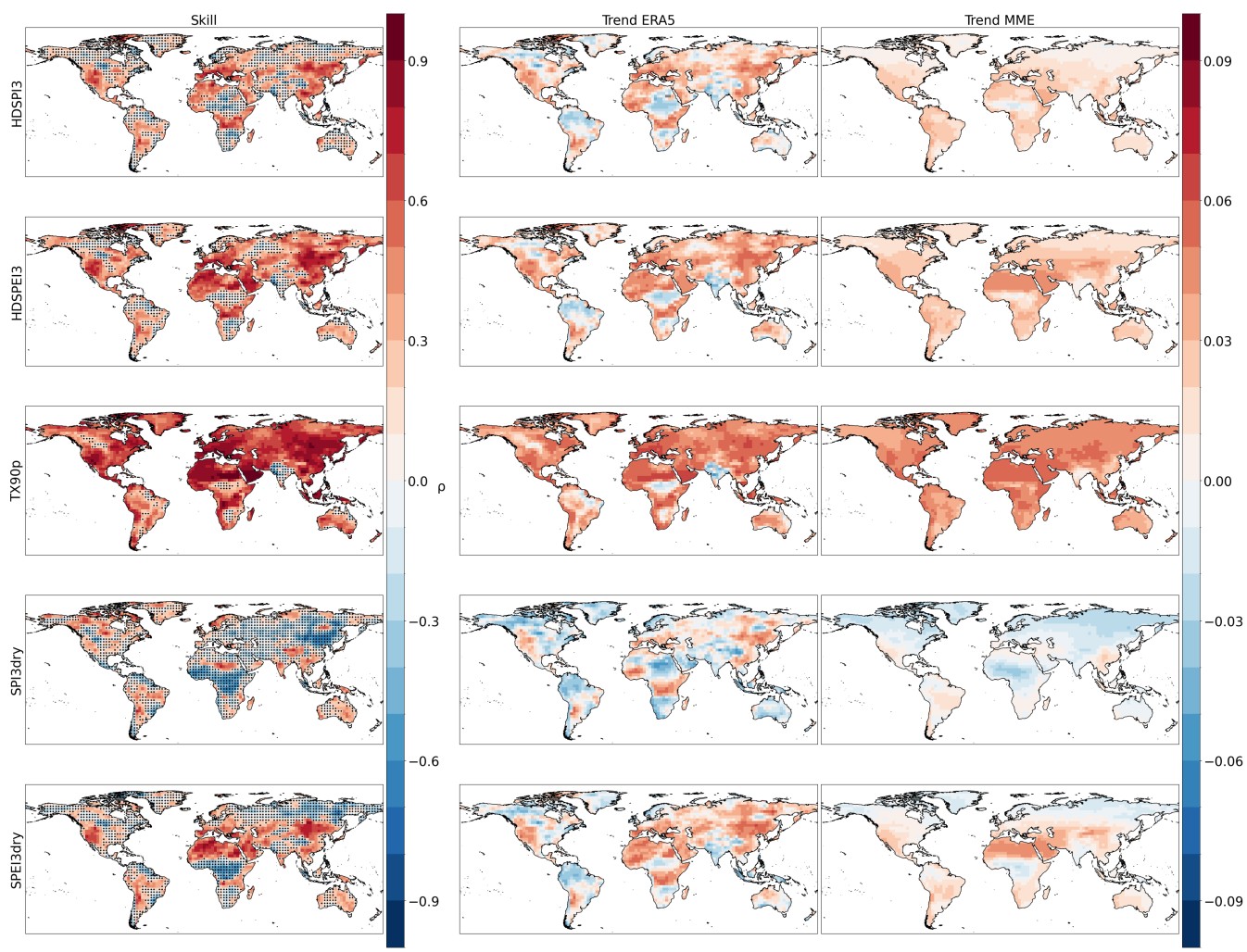

**Figure A5.** First row: Spearman rank correlation between the ERA5 dataset and the DCPP MME ensemble mean for HDSPI3, HDSPEI3, TX90p, SPI3dry and SPEI3dry respectively. Hatchings indicate correlations not statistically significant at the 95% confidence level (p-value < 0.05) using a one-tailed test. Second row: Normalized trends in the ERA5 dataset for HDSPI3, HDSPEI3, TX90p, SPI3dry and SPEI3dry respectively. Third row: Median normalized trends, calculated ensemble-wise, in the DCPP MME ensemble for HDSPI3, HDSPEI3, TX90p, SPI3dry and SPEI3dry respectively.

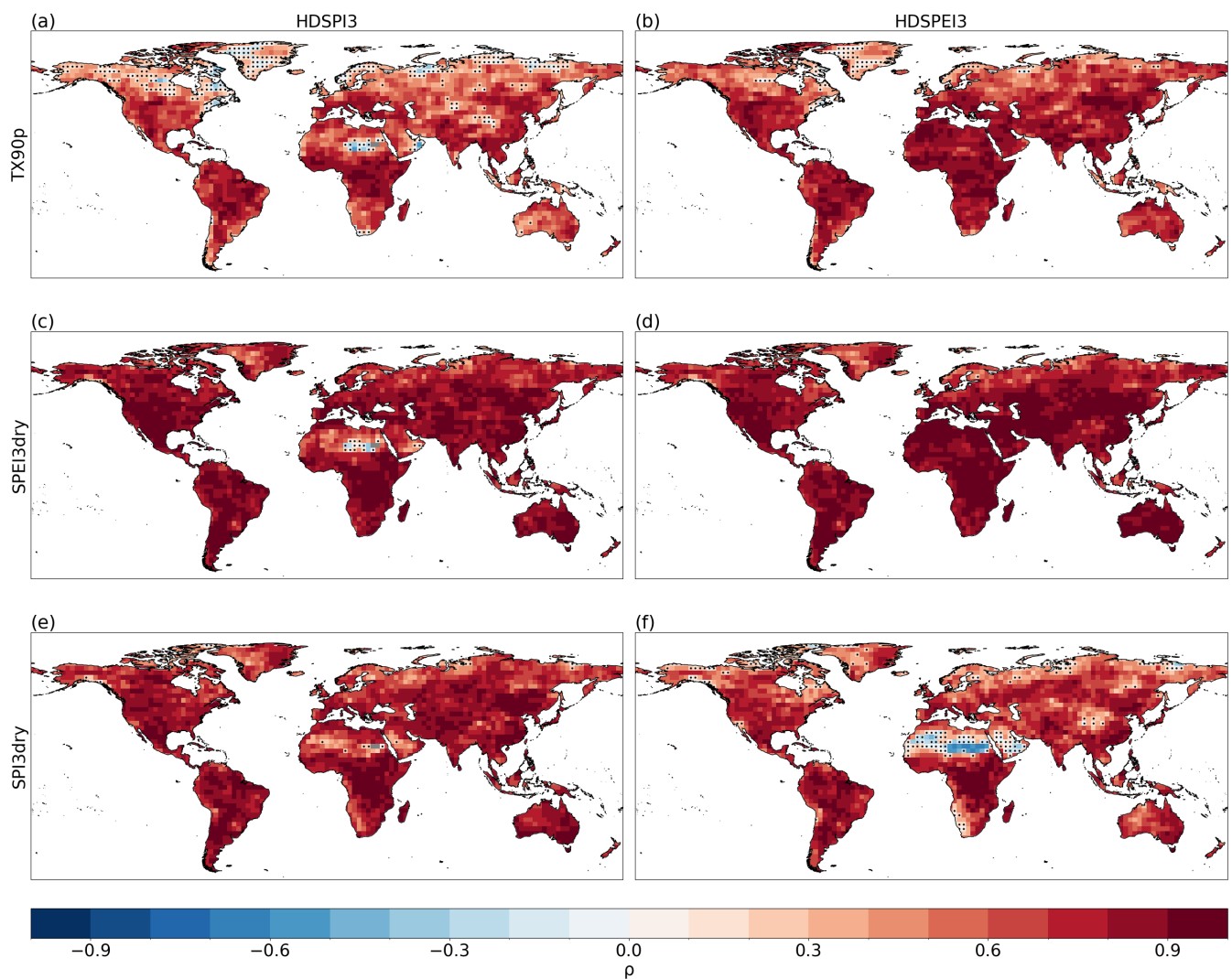

**Figure A6.** Spearman rank correlation between the annual sum of HDSPI3 (a,c,e) and HDSPEI3 (b,d,f) and TX90p, SPEI3dry and SPI3dry respectively, in the ERA5 dataset. Hatchings indicate correlations not statistically significant at the 95% confidence level (p-value < 0.05) using a two-tailed test.

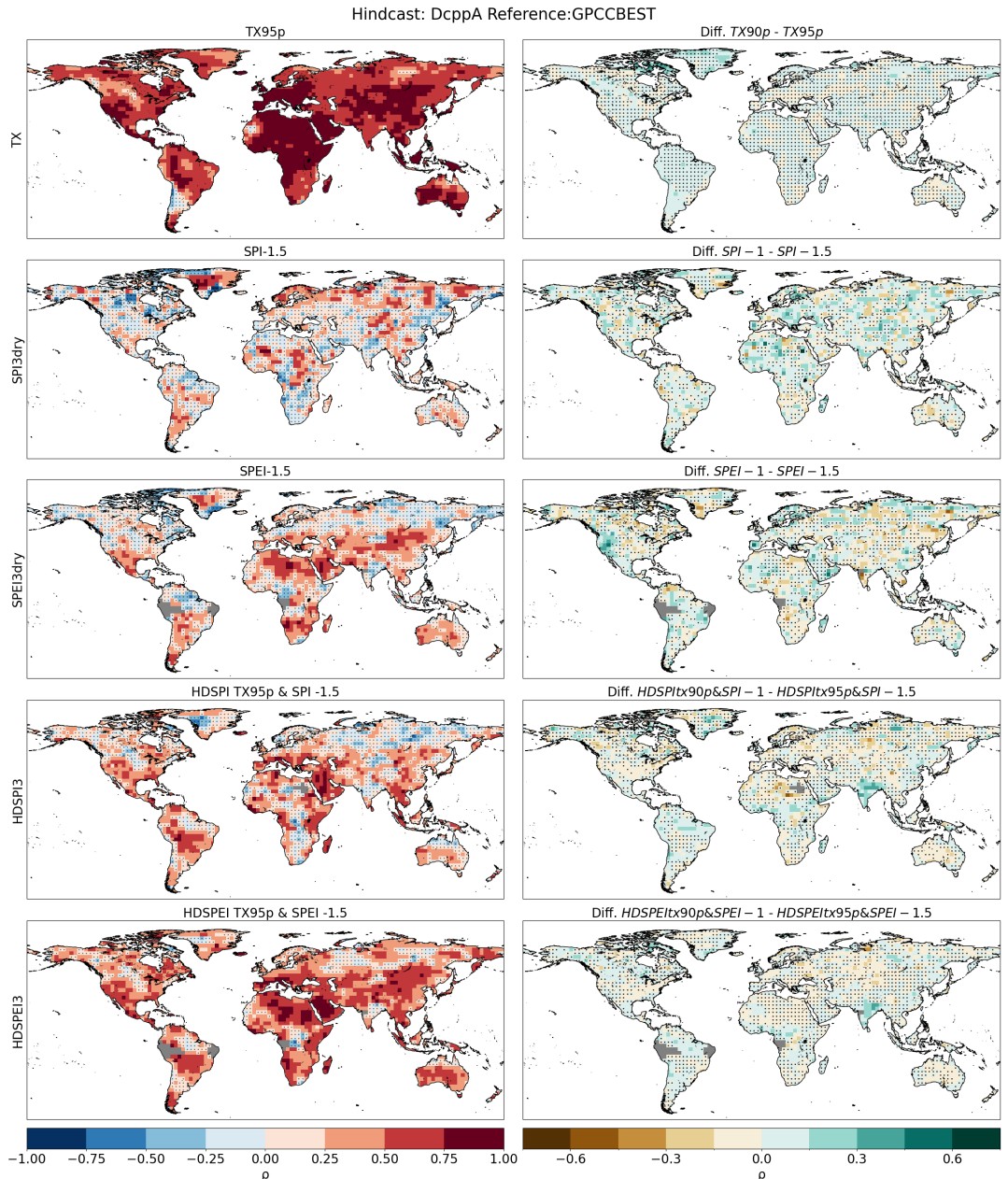

**Figure A7.** First column: Spearman's rank correlation for calculated for different extreme thresholds (First column). Correlations are performed between the DCPP MME ensemble mean and the GPCC-BEST dataset. Second column: Difference between the thresholds used in the main study and the ones shown in this figure. Hatchings indicate correlations (or differences between correlations) not statistically significant at the 95% confidence level. The significance of the difference has been computed using a Steiger Z-test. Grey areas indicate missing data.

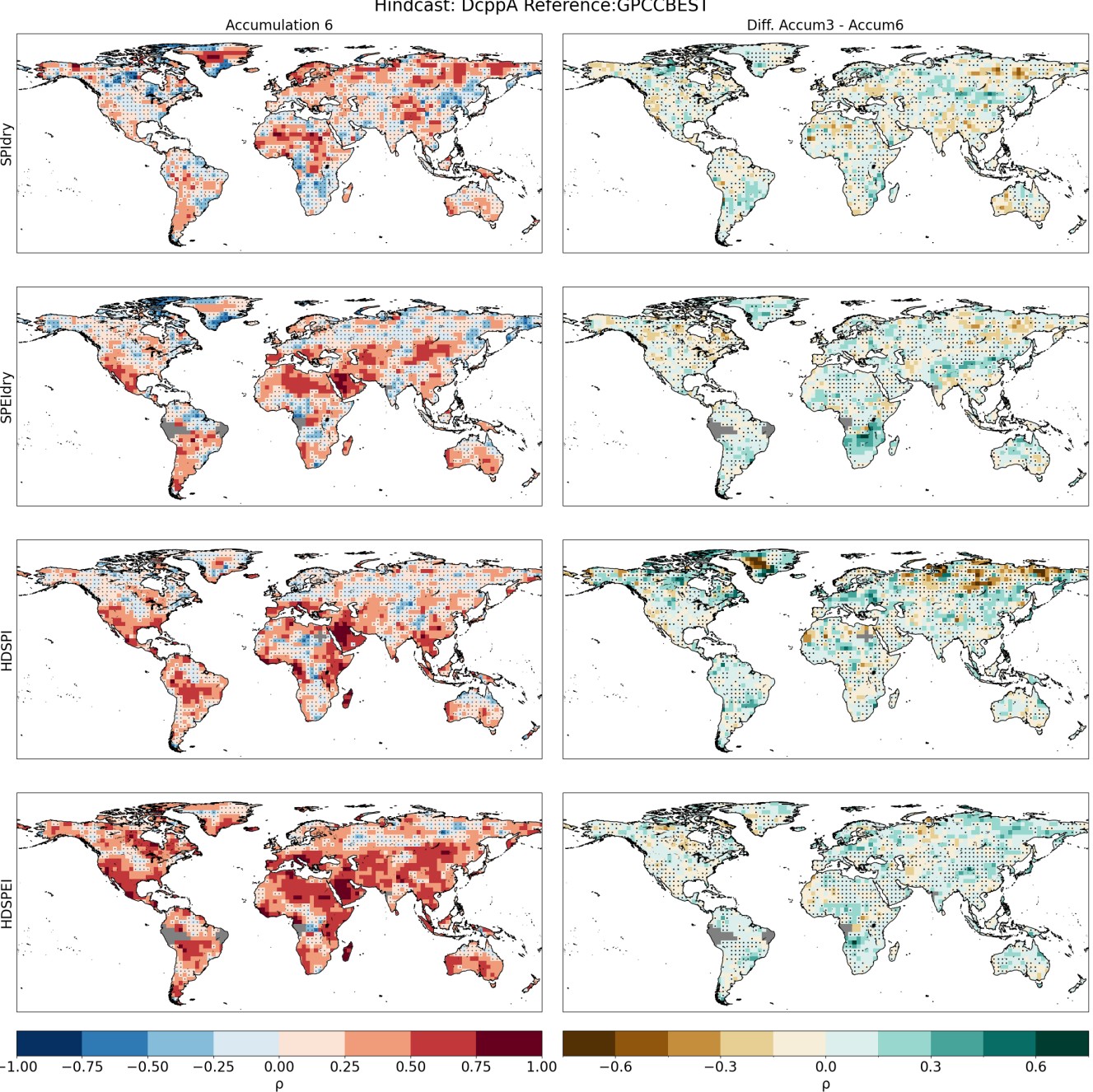

**Figure A8.** First column: Spearman's rank correlation for calculated for dry and hot-dry compounds with a 6-month accumulation. Correlations are performed between the DCPP MME ensemble mean and the GPCC-BEST dataset. Second column: Difference between the accumulation used in the main study and the one shown in this figure. Hatchings indicate correlations (or differences between correlations) not statistically significant at the 95% confidence level. The significance of the difference has been computed using a Steiger Z-test. Grey areas indicate missing data.

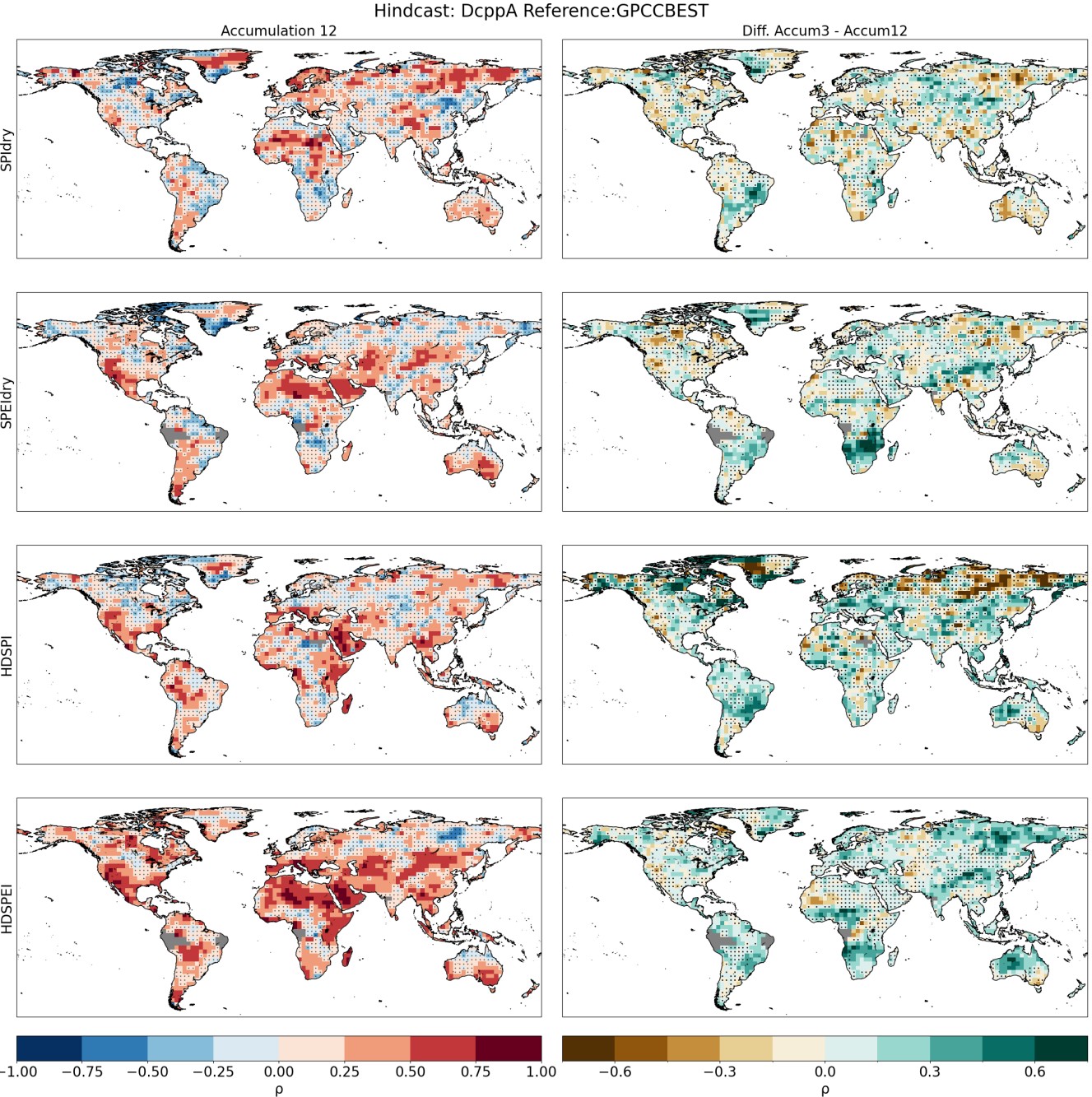

**Figure A9.** First column: Spearman's rank correlation calculated for dry and hot-dry compounds with a 12-month accumulation. Correlations are performed between the DCPP MME ensemble mean and the GPCC-BEST dataset. Second column: Difference between the accumulation used in the main study and the one shown in this figure. Hatchings indicate correlations (or differences between correlations) not statistically significant at the 95% confidence level. The significance of the difference has been computed using a Steiger Z-test. Grey areas indicate missing data.

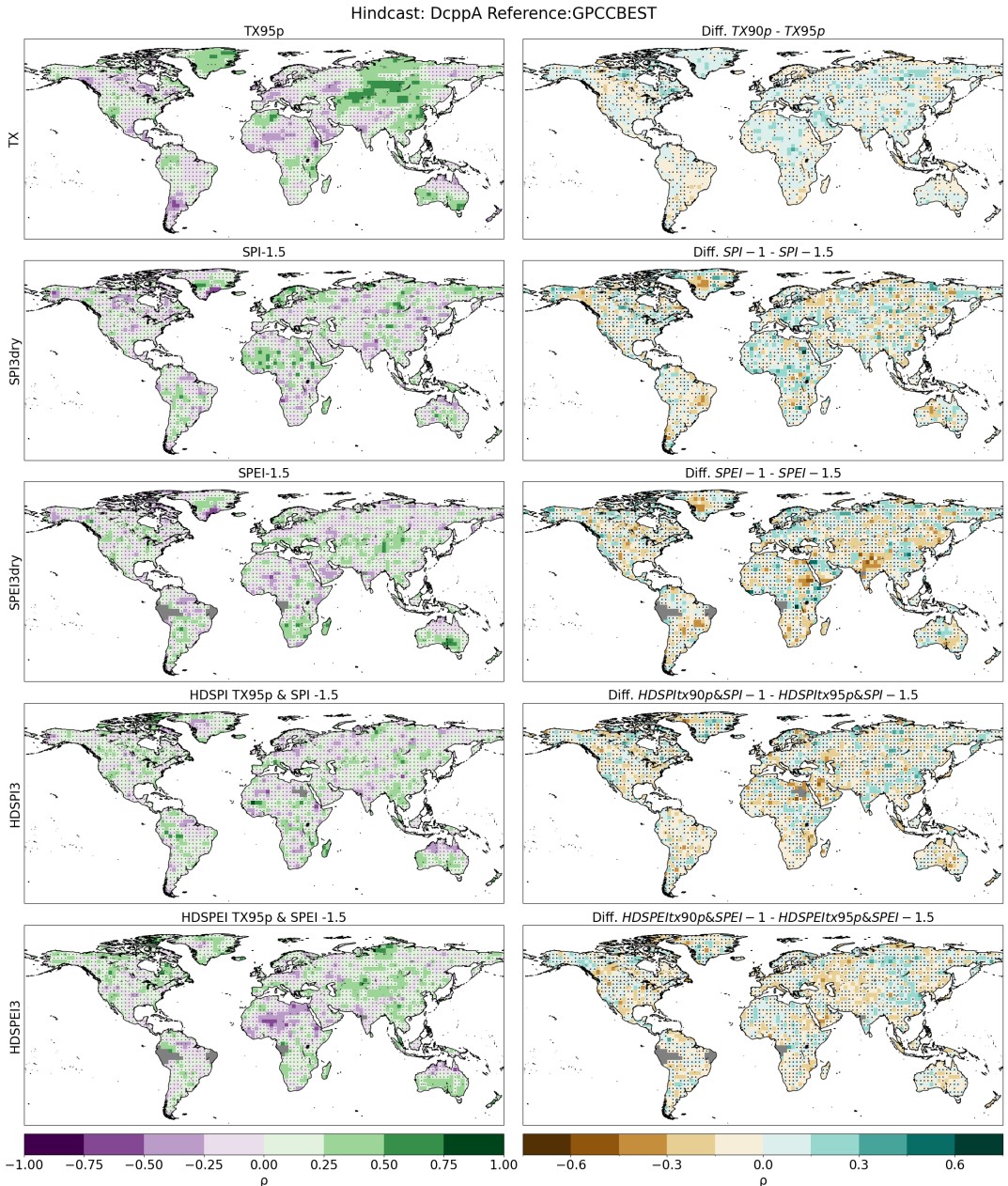

**Figure A10.** First column: Spearman's rank residual correlation calculated for different extreme thresholds. Correlations are performed between the DCPP MME ensemble mean and the GPCC-BEST dataset. Second column: Difference between the thresholds used in the main study and the ones shown in this figure. Hatchings indicate correlations (or differences between correlations) not statistically significant at the 95% confidence level. The significance of the difference has been computed using a Steiger Z-test. Grey areas indicate missing data.

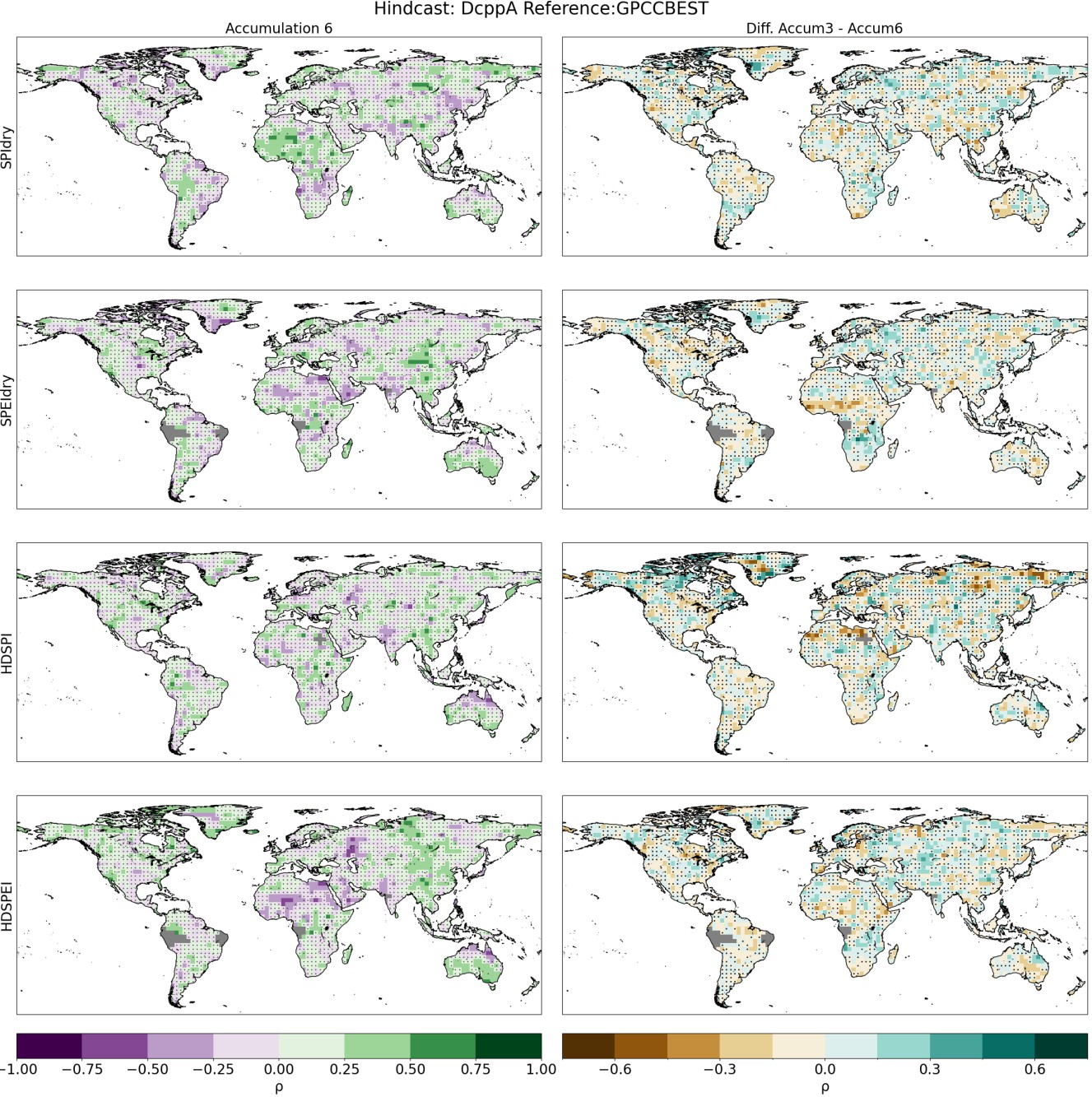

**Figure A11.** First column: Spearman's rank residual correlation calculated for dry and hot-dry compounds with a 6-month accumulation. Correlations are performed between the DCPP MME ensemble mean and the GPCC-BEST dataset. Second column: Difference between the accumulation used in the main study and the one shown in this figure. Hatchings indicate correlations (or differences between correlations) not statistically significant at the 95% confidence level. The significance of the difference has been computed using a Steiger Z-test. Grey areas indicate missing data.

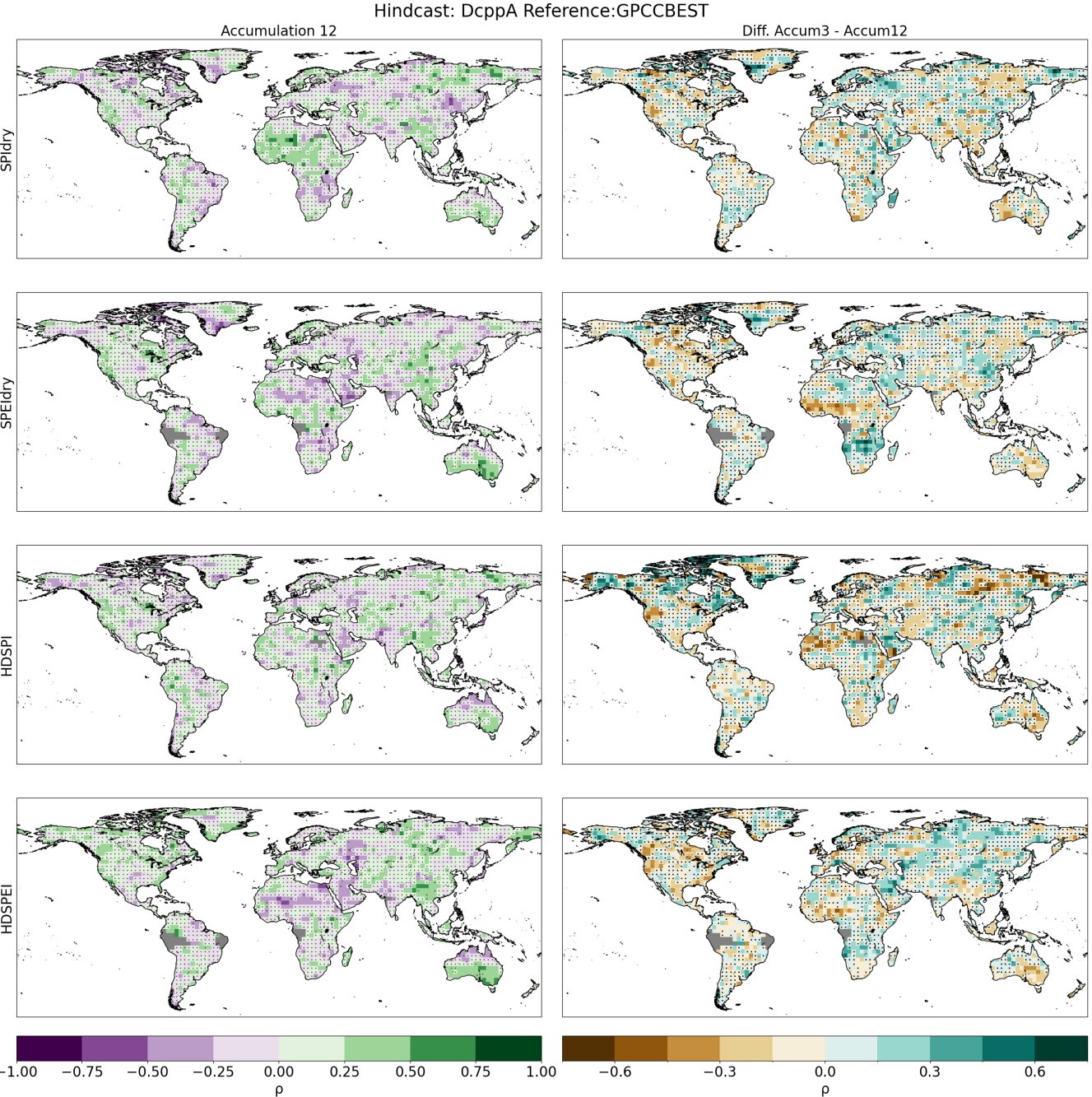

**Figure A12.** First column: Spearman's rank residual correlation calculated for dry and hot-dry compounds with a 6-month accumulation (First column). Correlations are performed between the DCPP MME ensemble mean and the GPCC-BEST dataset. Second column: difference between the accumulation used in the main study and the one shown in this figure. Hatchings indicate correlations (or differences between correlations) not statistically significant at the 95% confidence level. The significance of the difference has been computed using a Steiger Z-test. Grey areas indicate missing data.