# Peer review of "Multi-annual predictions of hot, dry and hot-dry compound extremes"

_EGUsphere, 2025_

## Author Comment (AC1)

Review: Multi-annual predictions of hot, dry and hot-dry compound extremes

The study by Aranyossy et al. investigates the predictability of hot-dry compound extremes and their univariate components (hot and dry events) on a multi-annual scale (forecast years 2–5), using decadal climate hindcasts from the CMIP6 Decadal Climate Prediction Project (DCPP). The study evaluates the skill of initialized forecasts compared to historical simulations, explores the relative contributions of external forcings versus initial conditions, and assesses whether the model ensemble can reproduce observed relationships between compound and univariate extremes.

This manuscript addresses a timely and scientifically relevant topic—the decadal predictability of compound hot-dry extremes—and provides valuable insights using a multi-model ensemble of CMIP6 decadal forecasts. However, the current version has some substantial shortcomings that limit its scientific clarity and impact. I believe the study requires major revision to strengthen the methodological framework, sharpen the interpretation of key results, and improve its overall scientific accuracy before it can be considered for publication.

We thank the reviewer for taking the time to review our manuscript, and for the constructive feedback. Please see below your comments and our answers highlighted in red.

**Major Notes:**

Unclear Justification for Compound Event Definition: The definition of hot-dry compound extremes is based on overlapping thresholds (TX90p and SPI3dry or SPEI3dry), but the manuscript does not adequately discuss how sensitive the results are to these thresholds or to the chosen accumulation window. The absence of sensitivity analysis raises questions about the robustness of the results.

In response to the reviewer's comment, we have performed a comprehensive sensitivity analysis using different thresholds and included the results as supplementary information. For hot extremes, we use an alternative threshold of the 95th percentile of the maximum temperature (TX95p). For dry extremes, we select accumulated months from SPI3 and SPEI3 with values <= -1.5 (SPI3-1.5 and SPEI3-1.5). Hot-dry compound extremes have also been calculated with the new thresholds. Additionally, we have performed a sensitivity analysis for 6-month and 12-month accumulation periods in the calculation of SPI and SPEI. The results have been added to the Appendix (Figures A7-A12). Based on the results of this analysis, we have observed that a change in the threshold does not yield significant differences. On the other hand, a change in the accumulation period reveals a corresponding change in correlations, particularly in specific regions such as Northern Asia and Southern Africa. We have added a paragraph in Sections 3.1 (line 174) and 3.2 (line 206) comparing the different thresholds and accumulation to the correlation and residual correlation, respectively. In Section 4 (line 329), we have added a paragraph summarising and discussing the overall results of the sensitivity analyses.

Overstatement of Skill Based on Trend Agreement: The study repeatedly refers to "skillful prediction" of compound extremes, but a substantial portion of this skill stems from long-term trend agreement rather than the successful prediction of interannual variability. In many regions, the DCPP ensemble appears to simply capture externally forced warming trends, which correlate with observed trends in hot extremes. However, this does not necessarily equate to predictive skill in a practical, decision-relevant sense. This distinction is mentioned, but not emphasized

sufficiently in the framing of the results or the conclusion. The authors must clearly distinguish between correlation due to trend matching and actual initialized predictive skill.

We agree with the reviewer's comment, and in response, we have attempted to better highlight the implications that a lack of residual correlation brings to overall results. Specifically, in Section 3.2 (line 250), we have added a sentence highlighting how the results of that section underscore the limitations of the current predictive potential of multi-annual predictions for hot-dry compound extremes. In addition, we have added a paragraph in the discussion Section (Section 4, lines 310-316) that underlines the distinction between forcing-derived and initialisation-derived skill, and its impact on the usability of these forecasts.

Presentation and Clarity: The text is dense and often difficult to follow due to inconsistent terminology and lengthy, complex sentences. Key methodological steps are underexplained or relegated to figure captions.

In response to the reviewer's comment, we have proofread the manuscript, with a particular focus on the delivery and clarity of the text. Specifically, we have homogenised the terminology used (Example: referring to hot-dry compound extremes throughout the methodology, results and discussion, instead of using several expressions such as events, extremes... we use only "hot-dry compound extremes"; we use the term "prediction" for the decadal predictions instead of referring to them as forecast, predictions, products...). In addition, we have shortened several otherwise lengthy sentences throughout the manuscript (Example: Section 1, lines 18-20, the sentence "The combination of . . . severe underestimation of the risk" was split into two sentences; in Section 2, lines 36-39, the sentence "Unlike climate projections . . . model initialisation" was split into two). Finally, we have also added some explanations, which were previously in the captions to the main text (Example: in section 3.2, lines 219-223, we have added a paragraph explaining Figure 3 and how we represented the connection between significant correlations and agreements of the trends' sign.)

**Minor Notes:**

Figure captions could benefit from clearer labeling and direct interpretation; currently, they are overly technical.

To integrate the reviewer's comments into the manuscript, we have reduced the amount of technical information in the captions, especially that which is already provided elsewhere, thus avoiding repetition between the text and captions. For example, in Figure 3 we have removed the explanation of the correlation-trend's sign and added a paragraph in the text. We have removed from the Figures the information stating the two datasets involved. For example, the sentence "Correlations are performed between DCPP MME and GPCC-BEST" has been removed, at least in the main paper, since in the Appendix, where we present the results for ERA5, we believe it is better to specify the dataset. Additionally, we have removed the part in the captions stating the alpha value, as well as the part where we specify whether we used a two-tailed or one-tailed test, since this information is already specified in the methods. Finally, in the Appendix Section, specifically in Figures A4 and A5, we also removed repetitions of the variables shown in the Figures.

l.87 add "as" (. . . we define months with drought conditions as all months . . . )
Added.

Several commas are missing throughout the text.

While proofreading the text, we have added the missing commas.

Geographical areas are not always written in the same way (for example, word "northern"
is sometimes written in lower case and sometimes in upper case).

To homogenise the text, we have checked the text and corrected the typos, following the rule
where "northern/southern", when used as adjectives, are written in lower case ("southern part
of. . . "), while when part of a proper noun, they are written in upper case (Example: North
Africa, South America, Northern Hemisphere).

l.238 delete "seen"
Deleted.

---

## Author Comment (AC2)

Foremost, I would like to mention, that researching compound extremes in the context of decadal climate prediction is a valuable contribution to the field of inter-annual predictions. The paper does represent the added value of initialized climate predictions well, but also shows the limitation of added value to specific variables and regions.

We thank the reviewer for taking the time to review our manuscript and provide helpful suggestions. Please see below your comments and our answers highlighted in red.

**Typos:**
4 - „IN this regard"
Amended.

145 - circumscribed - is there a better word possible, or just leave it out?
In response to the reviewer's comment, the sentence "are circumscribed and isolated" has been replaced with "areas are scattered and spatially isolated".

**General Remarks:**
0 - Is the code with which the results are produced available publicly?
To provide public access to the code, as well as the datasets used in this study, we have added a "Data Availability Statement". Here, we provide links to the public access of the datasets used in this study (CMIP6 models, GPCC, BEST, and ERA5 datasets), as well as a link to a GitLab project where the scripts used to compute the univariate and hot-dry compound extremes can be found (`https://earth.bsc.es/gitlab/aaranyos/hotdry_compounds_dcppa`).

70 - Did you apply any bias correction/calibration to the MME, individual models respectively? If not, why? Wouldn't a non-linear calibration add value to the model outputs?
Decadal predictions can be affected by model drift, which is typically corrected for by calculating anomalies relative to a lead-time dependent climatology. Here, we use a lead-time dependent percentile to calculate the temperature extremes. Specifically, we build a distribution for every lead-day of the predictions (with its 5-day window). On the other hand, for the standardisation of SPI and SPEI, the distribution is built for every lead-month. These specific steps for calculating the extremes, being lead-time dependent, implicitly correct for the drift in the decadal predictions. To better clarify this point in the manuscript, we have added, in Section 2.1, two brief sentences explaining these concepts, specifically at line 85 for the hot extremes, and line 105 for the dry extremes.

107 - the authors focus their study on lead times 2-5 which is in my opinion a fair choice. Could you give a reason on why you specifically choose this lead-time? And, if available, please present the findings for other lead times in the way you did for the comparison between GPCC-BEST/ERA5. Decadal predictions extend out to ten years, so I would be curious what happens to the signal of the initialization in this MME context.
The choice for the lead-years 2-5 was taken due to previous consultation of skill assessments of decadal predictions, which indicate the first lead-5 years to have a greater impact from the initialization. To help highlight this point better, we have added a sentence in section 2 (line 67), where we explain how the effect of initialization on this subset of decadal forecasts makes

them a desirable timeframe, especially for the skill assessment of extreme events. We also refer to Section 1 (lines 44-49), where we explicitly state how the first years of decadal forecasts show the most promise in terms of predictive skill. We have not performed the analysis of this study on the whole decadal period, but we would expect a slight degradation of the skill in longer lead times, especially regarding the contribution from initialization. We also refer to the study from Delgado-Torres et al., (2023) "Multi-annual predictions of the frequency and intensity of daily temperature and precipitation extremes", where in addition of the 2-5 lead years in the main study, in the Supplementary Material figures are shown also for the skill of the whole 10 years. The results show small differences compared to the 2-5 lead-year period, expect a degradation in the residual skill.

135 - In a production scenario, if you only had time to use one reference data set, would you recommend to use gridded observations or reanalysis?
Skill evaluations are always done for hindcasts, which for all decadal prediction systems are produced before any production or (operational) forecasts. This allows to evaluate the hindcasts against all available observational datasets, and in fact, it is important to take potential uncertainty related to the observational reference product into account; we have added a brief discussion on these uncertainties in section 4 (line 326). The forecasts as such would not depend on the reference dataset used for hindcast evaluation. For these reasons, we do not recommend one over the other.

149 - Given the interpretation of skill occurring in certain regions is difficult, I'm curious why Greenland and "Central Asia" stand out as showing increased skill due to initialization. Do you have any idea why that might be?
We agree that investigating the sources of such strong residual skill in hot extremes would be an interesting contribution to the topic. However, at this point we could only speculate about the reasons for the added skill. For example, the strong residual correlations found in Greenland could be related in some way to the decadal predictability of North Atlantic blockings, as stated in the study by Athaniasidis et al. (2020), *"Decadal predictability of North Atlantic blocking and the NAO"*. On the other hand, the residual correlation in Central Asia may be linked to the Siberian High, a relatively stable high-pressure system in the region. However, a more conclusive and detailed analysis of the sources of the skill should be done in follow-up studies. For this reason, we did not add speculations in the manuscript at this point.

---

## Author Comment (AC3)

This paper motivates that the assessment of decadal climate predictions is essential to providing reliable information on hot-dry compound extremes because of their potential impacts on environments and societies. There is mention that previous research has focused mostly on climate variables or univariate climate extreme prediction at this timescale and this study aims to fill that gap for hot-dry compound extremes. It evaluates the ability of the CMIP6 multi-model decadal climate hindcasts in predicting hot-dry climate extremes as well as hot and dry univariate counterparts for forecast years 2-5, investigates the added value of model initialization by comparing the forecasts to historical simulations, and compares the modeled correlations between compound and univariate extremes with observation-based datasets, ERA5 and GCPP-BEST (as referred to in the paper).

While the paper addresses a relevant gap and performs a worthwhile correlational analysis, much revision is required to make this study well-explained and well-interpreted.

We thank the reviewer for taking the time to review the paper and post the public comment. We did our best to take your comments into account and improve our manuscript. Please see below our answers.

**Major Points – Scientific Methods**
Strengths:
Two different CMIP6 experiment types (DCPP MME and Hist MME) are a meaningful way to explore the skill contributions for initialization and external forcing.

Good choice to use multiple reference datasets (GPCC-BEST and ERA5).

Critiques:
Model initialization details should be discussed (e.g. initialized over land/sea). - Model selection and some information/justification concerning the models should be mentioned instead of simply referring to the appendix table.
Model initialisation in the DCPP project is not standardized. For this reason, we refer to the reference in the appendix table for each single model, as each one of them applies different methods and data. We have added a sentence in section 2 (line 63) referring to this issue.

More discussion about the potential oversimplification due to all of the ensemble averaging should be discussed.
Please note that the extremes are calculated for each ensemble member individually, without averaging. In climate prediction, the ensemble mean is typically interpreted as the predictable component of the signal. We follow the same approach in this study, and to make predictions, we average the member-specific extremes to derive the predictions. There are well-known issues that the magnitude of variations in the ensemble mean is smaller than the observed variations, a direct consequence of averaging the noise component across individual members.

"Observational uncertainty" is only quoted however, discussion is required to compare the general higher agreement with GPCC-BEST in comparison to ERA5 at least.
Thank you for pointing this out. We have added a paragraph in section 4 (line 326) comparing the results between the observation and the reanalysis reference dataset.

The compound extremes definition is somewhat rigid and may miss more complex co-occurrences. There should be consideration given to the simultaneity of event dynamics.

We understand and agree that we computed the compound extremes based only on statistical properties, which are meaningful in this specific study. Computing the compound extremes based on dynamical characteristics may also be very relevant. However, in the context of this paper, this is out of scope, but could be investigated in future studies.

A 3 month accumulation window is chosen and is appropriate for meteorological drought but may not capture many other drought timescales. The sensitivity to accumulation period should be discussed.

We have performed a sensitivity analysis for a 6-month and 12-month accumulation period (as well as different extreme thresholds). The results have been added to the supplementary material (Figures S7-S12). We have added a paragraph in sections 3.1 (line 174) and 3.2 (line 206) comparing the different thresholds and accumulation to the correlation and residual correlation, respectively. In section 4 (line 329), we have added a paragraph on the overall sensitivity analyses.

**Major Points – Explanations**
Strengths:
Model-dependent variability is mentioned.

Critiques:
Explanation of indices calculation relies on references and is otherwise unclear/not justified (e.g. why use the analysis period as reference period for TX90p and are observation years used for the comparisons aligned with forecast years 2-5?). Should include some explanation similar to references (e.g. for SPI/SPEI calculation and standardization processes).

In section 2.2 (line 124), we have added a sentence explaining how the reference datasets are aligned with the average forecast years 2-5 for the correlation analysis. In addition, we refer to the calculation of the extremes to studies which are widely known and established in the literature. More specifically, TX90p is part of the ETCCDI suite of indices (we refer in the manuscript to the study of Zhang et al. (2011)). On the other hand, for SPI and SPEI, we refer to the studies McKee et al. (1993) and Vincente-Serrano et al. (2010). Finally, we use the entire time series as a reference period for TX90p to ensure consistency with the reference periods of SPI and SPEI.

Methodological details feel condensed or fragmented:
Mention that PET is calculated using the Hargreaves's method but no justification is provided. In section 2.1 (line 99), we have added a brief explanation of why the Hargreaves method was reasonable for this specific study.

Description of percentile estimation using a 5-day running window is noted but how seasons are handled should be discussed.

The running window is short enough not to incur in the seasonality biases that are common in the longer window periods. We have added a sentence in section 2 (line 84) explaining this concept, and also added a reference (Brunner et al., 2021).

Missing data mentioned in figure captions but not discussed.

We have added a sentence in section 2 (line 75), indicating how missing data are part of observational datasets.

A statement about why hot-wet compound extremes are not discussed would strengthen the focus of this analysis.
We appreciate the suggestion, and indeed an investigation of different types of compound extremes would be interesting to analyse. However, in this study we focused on this specific type. We believe that analysing also hot-wet compounds would broaden too much the spectre of this specific research (including the need to define indices that meaningfully reflect e.g. the health risks of hot and humid conditions), and suggest this would be best to do in future research.

There is mention in the introduction that dependence among univariate variables of a compound extreme can decrease the return period of such events but this is not discussed or mentioned again any further.
The mention of the return period has been removed.

**Major Points – Conclusions/Interpretation of Results**
A general comment about the statement of results: overall it reads a bit as a list, with a lack of sufficient meaningful connections made or real reasons explored for greater "skill" in particular regions compared to others.

Critiques:
The minimal added skill from initialization is important, if a real artifact, but under-discussed – what does this imply for the use of decadal forecasts in operational contexts? In section 3.2 (line 183), we have added a sentence explaining the usefulness of residual skill. In addition, at the end of the section, we have added a sentence explaining how that section underlines the limits of the current predictive potential of multi-annual predictions for hot-dry compound extremes. In addition, we have added in the discussion (section 4, lines 310-316) a paragraph underlining the distinction between forcing-derived and initialisation-derived skill, and its impacts on the usability of these forecasts.
The regional and seasonal variations are mentioned as future work but some exploration into this would give more content/meaning to the study. A preliminary investigation would be interesting and relevant given the spatial nature of the data and existing literature.
We agree that exploring seasonal and regional variations would be a significant contribution to the field in which this study is situated. However, we believe that this would broaden the focus of this study; therefore, we prefer to reserve it for a future investigation.

Statements about better/under-performing regions are made but connections between them and between areas for similar analyses should be explored. These really should be accompanied by considerations of the frequency and intensity of these extreme events in those areas (e.g. high skill in California important for multi-day droughts in the region).
At this point, we can only speculate regarding the sources of better/worse skills in specific regions. However, a more conclusive and detailed analysis of the sources of this skill in such regions should be conducted in follow-up studies.

**Recommendations for Further Discussion/Improvement**

Aforementioned justifications/clarifications should be included. Thank you for the suggestions provided. We have tried to include them as best as we can.

Schematic for event selection (compound index calculation) would enhance clarity. We appreciate the suggestion. However, we believe the compound index to be already quite schematic. In addition, the study of De Luca and Donat (2023) already offers a quite comprehensive explanation of this specific method.

Consideration of other compound event types such as humid heatwaves, lagged dependences (or explanation for why specifically these that were chosen) would expand the framework's applicability. As stated above, we believe that considering other types of compound extremes and dynamics could be a valuable contribution to the topic, but is out of the scope of the current study. We have added a comment in Section 4 (line 345) where this framework should be applied also to other compound indicators.

Mention and discussion of the socio-economic impacts since the motivation centers on "high-impact" extremes but is never discussed again. Linked the compound events to impact datasets could strengthen the relevance and make the closing statement make more sense. We agree that an exemplification of the impacts would strengthen the manuscript's motivation. We have added a sentence in Section 1 (line 24), where we put a specific example of the effects of the hot-dry summer in Russia in 2010.

Specific mentions of events should be made rather than "compound extremes during 2003, 2010, 2015 and 2018 in Europe stand as an example". We have extended the sentence in Section 1 (line 25-26), talking about the examples of compound events. In addition, we refer to the references added at the end of the sentence in Section 1.

Sensitivity analysis on SPI/SPEI accumulation periods could reveal the robustness of event definitions. See comment above regarding the SPI/SPEI accumulation periods.

Skillful areas should be connected to real examples to emphasize the importance of the system's performance. We appreciate the suggestion. However, the scope of this study is to assess the skill of the hindcast over the whole time period. Exploring the hindcast's ability to predict single events would broaden the scope of this study too far.

Similarly, discussion of how the lack of skill in extreme-prone areas is a major limitation (e.g. North Africa, where certain types of hot-dry events are more common). We discuss the limits of the skill, particularly from the perspectives of residual correlation and interannual variability. Specifically, we have added short paragraphs in Section 3.2 (line 250) and Section 4 (line 314) to highlight this limitation. However, exploring the sources of such a limitation would be, at this point, a speculation. We could conduct a more specific analysis on the topic in future studies.

**Minor Points:**
Figures are should be made larger and less crowded.
Thanks for the comment. However, we use a maximum of six panels per figure. We are also utilising the largest available space and the optimal panel configuration to display the figures. The plots for the correlation, residual correlation, and correlation-trend analysis are created to highlight the hot-dry compound extremes.

This is small but there are too many references to the appendix mostly in the discussion for the ERA5 results but not presented in the body.
We use ERA5 as a secondary dataset. However, we decided to show only one dataset in the main paper, especially because the two datasets (GPCC-BEST and ERA5) are mostly in agreement.

Bias correction and model drift should be noted. This is mentioned in one of the authors' other papers referred to in the Data and Methods section.
Decadal predictions can be affected by model drift, which is typically corrected for by calculating anomalies relative to a lead-time dependent climatology. Here, we use a lead-time dependent percentile to calculate the temperature extremes. Specifically, we build a distribution for every lead-day of the predictions (with its 5-day window). On the other hand, for the standardisation of SPI and SPEI, the distribution is built for every lead-month. These specific steps for calculating the extremes, being lead-time dependent, implicitly correct for the drift in the decadal predictions. To better clarify this point in the manuscript, we have added, in Section 2.1, two brief sentences explaining these concepts, specifically at line 85 for the hot extremes, and line 105 for the dry extremes.

**Grammatical/Written Structure:**
Generally clear, scientific tone and use of domain-specific terminology.

Good integration of citations.

Minor/spelling/grammatical errors.
We have proofread the manuscript, correcting all the typos, grammatical errors we encountered.
Incorrect figure references (e.g. "Figure 2d" instead of correct "Figure 2c").
The reference has been corrected, and also the other references have been checked for other potential mistakes.
Incorrect figure references (e.g. "Figure 2d" instead of correct "Figure 2c").
We have shortened several otherwise lengthy sentences throughout the manuscript (Example: Section 1, lines 18-20, the sentence "The combination of . . . severe underestimation of the risk" was split into two sentences; in Section 2, lines 36-39, the sentence "Unlike climate projections . . . model initialisation" was split into two).
Paragraph transitions need to be made smoother and often miss key points – topic sentences should be prominent with a clear message while more concluding sentences need to be added to complete the idea of the paragraph.
We have tried to make the transitions between the paragraphs, and also the different sections of the manuscript, more fluent.

---

## Author Response (AR3)

Dear authors,

Thank you for the excellent and comprehensive review. Included below are a few minor points that should be addressed before publication.

We thank the editor for the thorough and useful review. Please find the comments below, with our responses in red.

Both reviewers have requested a discussion of the sensitivity of the threshold choices. Thank you for the additional analyses. By including longer SPI time-scales, you are including other types of droughts. Please add a short discussion to this end (see for example information here: `https://drought.emergency.copernicus.eu/data/factsheets/factsheet_spi.pdf`).

Thank you for the useful comment. Indeed, it is important to underline the different nature of the droughts that we obtain using different accumulation periods, and also their different predictability. We have added a short discussion on the topic at the end of Section 4 (line 326), as well as two sentences in the method part (line 122), specifically describing how the 3-, 6- and 12-months accumulations describe different types of droughts with different impacts, as well as possible effects that the accumulation period has on the predictability of this kind of events.

In the lead time dependent calculation of the SPI - do you see any changes in the underlying distributions? We have computed the mean and the standard deviation of the distributions for the calculation of SPI and SPEI (accumulated precipitation and water budget), for every lead month which was included in the study. We have then plotted the results for the initial (lead year 2) and final (lead year 5) leadyear. We then proceeded to subtract the values of lead year 5 from the values of lead year 2, to highlight the differences between the different lead times. We show below as an example, the result for the model HadGEM3-GC31-MM. In general, we find that the differences between lead year 2 and lead year 5 are both model- and month-dependent. For example, as it can be seen in the figures below, most of the models show a positive difference for equatorial South America, except CanESM5, which shows negative differences. It also appears that stronger differences, both in mean and standard deviations, appear between the months of January and June. These differences can be attributable to a remaining effect of the drift in decadal predictions, despite computing the distributions on a lead time level. However, all of the results show low differences compared to the absolute values of the mean and the standard deviations (not shown here), except in limited regions or specific gridpoints. In Section 2.1 (line 107), we have added a paragraph explaining the results of these analyses.

[Figure]

Figure 1: *Difference between the means of the lead month distributions for lead year 2 and lead year 5 calculated from accumulated precipitation (SPI). Values in the titles indicate the minimum and the maximum values of these differences.*

[Figure]

*Figure 2. Difference between the precipitation (SPI) standard deviations of the lead month distributions for lead year 2 and lead year 5 calculated from accumulated precipitation (SPI). Values in the titles indicate the minimum and the maximum values of these differences.*

[Figure]

*Figure 3. Same as Figure 1, but for the water budget (SPEI).*

[Figure]

*Figure 3. Same as Figure 2, but for the water budget (SPEI).*

One reviewer asks about the focus on years 2 - 5. What are examples of the preventive

measures that decision makers can take on these lead times? Who will be using information with a 2 to 5 year lead-time? If you have some insight, it would be greatly appreciated.

We thank the reviewer for the comment. Climate information on the frequency and intensity of hot-dry compound extremes at multi-annual time scales can support decision-making processes in several sectors, such as agriculture, water management, energy and infrastructure. We have added the following sentences at the end of the second paragraph in Section 1 (line 33):

"For instance, multi-annual predictions of compound hot-dry extremes can inform strategic and preventive decisions across sectors. For instance, they can support agricultural planning such as irrigation investments, the multiplication of drought-resilient crop varieties, post-harvest management, and enhance the preparedness against pests and diseases (Delgado-Torres et al., 2025). Multi-annual climate information can also guide infrastructure and energy planning (Dunstone et al., 2022), including the enhancement of energy storage capacity or reinforcement of urban green areas to mitigate heat and dry stress, and anticipate impacts on society and environment."

Request 135 is not really answered. The reviewer is seeking guidance from you. No changes in the manuscript are needed, but a more informative response to the reviewer is needed if you can provide it.

Thank you for the comment. We have modified the answer to the reviewer,which can be found above in this document. We have tried to highlight how, especially before the production step, it is important to test the hindcast against several reference datasets, since both of them show different strengths and limitations. For this reason, we do not feel in the position of recommending one over the other.

Model overview table: if feasible, please summarize the most pertinent information on the model initialization procedures there.

Thank you for the comment. We have added a column to the Model overview table titled "Initialization scheme". There, when we were able to find the information, we summarized the initialization process of every experiment and, if available, the main datasets used in the assimilation.

Reply to the comment on ensemble averaging: please add your reply and a reference to the manuscript.

Thank you for the comment. We have added in section 2.2 (line 128) a summary of the reply provided to the reviewer. Specifically, we stated that, while the computation of the indicators was done at an ensemble-member level, we used the ensemble mean in the forecast quality assessment as it is interpreted as the representation of the predictable signal. In addition, adding a couple of references on the topic (Eade et al., 2014; Smith et al., 2019).

Remark on the use of high-impact events: I recommend a reformulation along the lines that some of these events are high-impact events, but not all of them.

Thank you for the comment. Indeed, we agree that it is an important difference. In addition, the focus of this manuscript is not impacts, but rather the predictability of the hazards. For this reason, in section 1 (line 15), we have substituted "high impact weather events" with "extreme weather events".

Please increase the figure panel size as requested by the reviewer. The figure details are unreadable for not-so-young eyes.

The figures have been enlarged as suggested in the comment. More specifically, Figures 1,2 and 3 in the main text, as well as A1, A2 and A3 in the Supplementary Materials have been modified to have all the panels the same size. In addition, for figures A4 and A5 in the Supplementary Material, the figures have been reorganised vertically to enlarge the size of the panels.